

# Entropy driven inductive response of topological insulators

A. Mert Bozkurt[1,2,3*], Sofie Kölling[4],
Alexander Brinkman[4] and İnanç Adagideli[3,4,5†]

**1** QuTech, Delft University of Technology, The Netherlands
**2** Kavli Institute of Nanoscience,
Delft University of Technology, Netherlands
**3** Faculty of Engineering and Natural Sciences, Sabanci University, Turkey
**4** MESA+ Institute for Nanotechnology, University of Twente, Netherlands
**5** TÜBİTAK Research Institute for Fundamental Sciences, Turkey

⋆ a.mertbozkurt@gmail.com , † adagideli@sabanciuniv.edu

## Abstract

3D topological insulators are characterized by an insulating bulk and extended surface states exhibiting a helical spin texture. In this work, we investigate the hyperfine interaction between the spin-charge coupled transport of electrons and the nuclear spins in these surface states. Previous work has predicted that in the quantum spin Hall insulator phase, work can be extracted from a bath of polarized nuclear spins as a resource [1]. We employ nonequilibrium Green's function analysis to show that a similar effect exists on the surface of a 3D topological insulator, albeit rescaled by the ratio between electronic mean free path and device length. The induced current due to thermal relaxation of polarized nuclear spins has an inductive nature. We emphasize the inductive response by rewriting the current-voltage relation in harmonic response as a lumped element model containing two parallel resistors and an inductor. In a low-frequency analysis, a universal inductance value emerges that is only dependent on the device's aspect ratio. This scaling offers a means of miniaturizing inductive circuit elements. An efficiency estimate follows from comparing the spin-flip induced current to the Ohmic contribution. The inductive effect is most prominent in topological insulators which have a large number of spinful nuclei per coherent segment, of which the volume is given by the mean free path length, Fermi wavelength and penetration depth of the surface state.

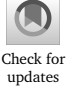

# 1 Introduction

After the discovery of quantum spin Hall insulators, the existence of a three dimensional version of the time-reversal invariant topological insulators was predicted [2,3]. 3D topological insulators (3DTIs) have a bulk band gap and host conducting states on their surfaces. These topological surface states (TSS) have perfect spin-momentum locking, i.e. the momentum of an electron is always perpendicular to its spin. An immediate consequence of perfect spin-momentum locking is the absence of backscattering for the topological surface states: similar to helical edge states of a quantum spin Hall insulator, topological surface states of 3DTIs do not suffer from localization under any time-reversal invariant perturbation, provided that the bulk band gap is not closed. Although backscattering is absent, the scattering probability in other directions is finite. Therefore, transport on the surface of a disordered 3DTI can be diffusive, whereas transport remains ballistic for the edge states of quantum spin Hall insulators even in the presence of nonmagnetic disorder. The diffusive limit of the 3D topological insulators and its transport properties have been studied in detail [4–6], and several materials have been verified experimentally [7].

Another prominent feature of spin-momentum locking is the Edelstein effect [8], a phenomenon previously observed in semiconductors featuring Rashba spin-orbit interactions [9–12]. For the surface states of a 3DTI, this phenomenon leads to the generation and control of spin accumulation along the surface of a 3DTI in an electric field [13–15]. This concept has profound implications for the field of spintronics, where the manipulation of electron spin for information storage and processing holds immense promise [16]. Notably, experimental demonstrations of the Edelstein effect and the inverse Edelstein effect in 3DTI materials have been reported [17,18], showcasing the feasibility of generating and detecting spin accumulation by applying electric fields in 3DTIs.

The Edelstein effect also leads to dynamic nuclear spin polarization (DNSP) [19,20]. This effect involves the transfer of nonequilibrium spin accumulation from charge carriers to nuclear spins. However, the reverse process is also possible: finite nuclear spin polarization can

generate a nonequilibrium electron spin accumulation. This phenomenon, facilitated by spin-momentum locking of the charge carriers in 3DTIs (see Fig. 1(a)), drives a charge current through the inverse Edelstein effect. This interplay between nuclear and electron spins has garnered attention for its potential to realize a Maxwell's demon effect in quantum spin Hall insulators [1] and quantum anomalous Hall insulators [21].

In this manuscript, we investigate the effect of nuclear spins on the topological surface states of a 3DTI, within the framework of a transport setup shown in Fig. 1(b). Combined with the hyperfine interaction between nuclear spins and electrons, we show that the Edelstein effect in a topological surface state can lead to DNSP. In return, we find that finite nuclear spin polarization effectively induces a charge current response within the system through the mechanism of the inverse Edelstein effect. Unlike the helical edge states of 2D topological insulators, topological surface states of a 3DTI can also scatter from nonmagnetic impurities (see Figure 1(c-d)), leading to a diffusive transport regime. The presence of this additional source of scattering enriches the complexity of the problem, making it more interesting to investigate theoretically.

Furthermore, we find that the charge current induced by nuclear spin-flip interactions is of inductive nature, akin to the predicted effect in quantum spin Hall edge states. The efficiency of the inductive power generation is characterized by a quality factor, denoting the ratio between reactance and resistance of the topological surface state. We show that this quality factor is enhanced by maximizing the amount of nuclear spins in-between impurity scattering events. At low frequencies, the inductive effect reduces to a universal inductance value, which is scalable by altering the aspect ratio of the surface of a 3D topological insulator. This enables miniaturization of inductive circuit elements. Finally, we estimate the induced current in a few exemplary materials, providing a framework for experimental applications.

This manuscript is organized as follows: In Section 2, we employ nonequilibrium Green's function formalism to describe the electron dynamics at the surface of a 3DTI, taking into account both interactions with nuclear spins and nonmagnetic impurities. This leads to a current-voltage relation, where an Ohmic contribution is accompanied by an induced charge current, set by the nuclear spin-flip rate. Section 3 builds upon this result, by finding an equivalent electronic circuit configuration, consisting of two resistors and an inductor, that describes the topological surface state including the spin-flip interaction with the nuclear spins. In Section 4, we relate the power delivered to and generated by the topological surface state to the entropy of the nuclear spin subsystem, and find that the inductive response has an entropic nature. Finally, in Section 5, we provide an overview and conclusion.

## 2 Diffusion at the surface of a 3D topological insulator

### 2.1 The model

We start with a low-energy effective Hamiltonian that describes the topological surface states of a 3DTI. Without loss of generality, we focus on the top surface of a 3DTI with a single Dirac cone:

$$H_0 = \hbar v_F (\boldsymbol{k} \times \boldsymbol{\sigma}) \cdot \hat{z}, \qquad (1)$$

where $v_F$ is the Fermi velocity of the Dirac fermions, $\boldsymbol{k} = (k_x, k_y)$ is the momentum operator, $\boldsymbol{\sigma}$ are the Pauli matrices that describe the spin degree of freedom of the charge carriers. As shown in Fig. 1(a), the dispersion relation derived from Eq. (1) features perfect spin-momentum locking.

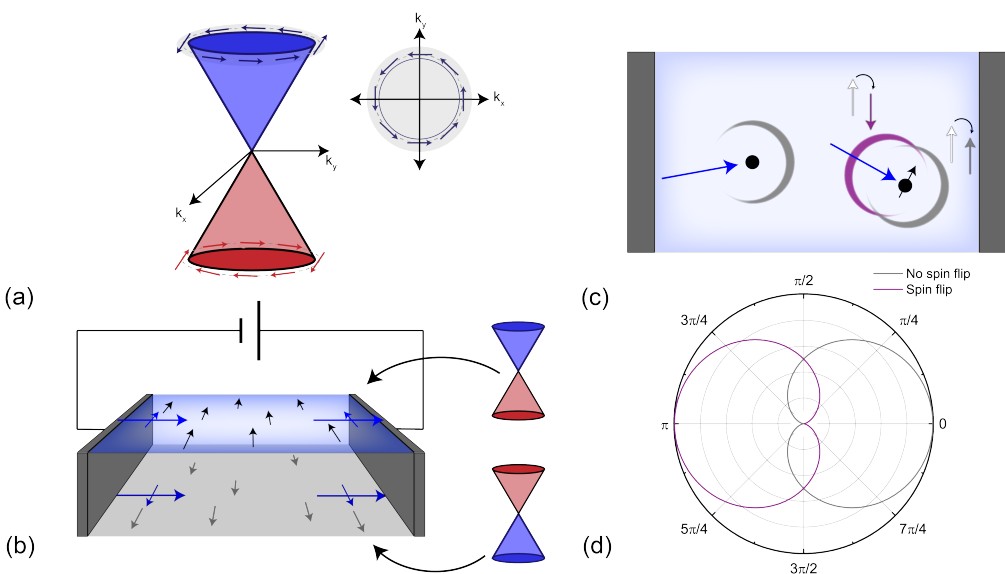

Figure 1: (a) Dispersion relation of a topological surface state, with spin being locked perpendicular to the momentum direction. (b) Schematic description of the transport setup involving the top surface of a 3DTI connected to metallic leads. In response to a charge current flowing through each surface, the charge carriers for top and bottom surfaces are spin-polarized in the opposite direction. Nuclear spins (black/grey arrows) on the top/bottom surface are polarized through spin-flip interactions with spin-polarized charge carriers. (c) Scattering processes at the top surface of a 3DTI. The incoming electron is depicted as a blue arrow, the normal impurity is depicted as a black dot and a magnetic impurity/nuclear spin is depicted as a black dot with an arrow. The normal scattering process is depicted with a grey arc and nuclear spin scattering is depicted with a purple arc. In each case, the arc thickness corresponds to the angle-dependent scattering probability. Depending on whether a spin-flip interaction takes place, forward or backward scattering is enhanced or suppressed. (d) Scattering probability as function of the scattering angle, for processes including (purple) or excluding a spin flip (grey). This scattering probability corresponds to the arc thickness in (c).

In this context, we consider two sources of scattering for topological surface states: nonmagnetic impurities and nuclear spins. We consider a nonmagnetic impurity potential $V(\boldsymbol{x})$, specified by a random Gaussian disorder profile $\langle V(\mathbf{x})V(\mathbf{x}')\rangle = n_0 U^2 \delta(\mathbf{x}-\mathbf{x}')$ and zero mean value $\langle V(\mathbf{x})\rangle = 0$. Here, $U$ is the magnitude of the nonmagnetic impurity scattering strength and $n_0$ is the nonmagnetic impurity density. Finally, $H_{hf}$ is the hyperfine interaction between the electrons and the nuclear spins:

$$H_{hf} = \frac{\lambda}{2}\sum_n \mathbf{I}^n \cdot \boldsymbol{\sigma}\,\delta(\mathbf{r}-\mathbf{r}_n).\qquad(2)$$

Here, $\lambda = A_0 v_0/\xi$ is the effective hyperfine interaction strength between electrons and nuclear spins, $A_0$ is the hyperfine coupling, $v_0 = a_0^3$ is the unit cell volume, $\xi$ is the surface state decay length in $z$-direction. $\mathbf{I}_n$ represents the Pauli spin matrices for the $n^{\text{th}}$ nuclear spin residing on the surface at position $\mathbf{x}_n = (x_n, y_n)$.

## 2.2 Nonequilibrium Green's function of the topological surface states

To describe the dynamics of the electrons interacting with both nuclear spins and nonmagnetic impurities, we employ the nonequilibrium Green's function formalism [22]. The structure of the perturbation expansion for the nonequilibrium Green's function is similar to the equilibrium theory, with the only difference being the introduction of a contour. The diagrammatic formulation of the Keldysh technique is almost identical to the equilibrium diagrammatic formulation, except for the fact that the propagators and vertices contain contour indices [22].

In derivation for the transport equation for the electrons, the central quantity of interest is the electronic Keldysh Green's function:

$$\underline{G}(1,1') = \begin{bmatrix} G^R(1,1') & G^K(1,1') \\ 0 & G^A(1,1') \end{bmatrix}, \tag{3}$$

where we use the abbreviation $1 \equiv (\boldsymbol{x}_1, t_1)$. We note that the matrix elements of the Green's function $\underline{G}$ are also matrices in spin space for the specific problem we consider. The diagonal elements of $\underline{G}$, namely $G^R$ and $G^A$, are the retarded and advanced Green's functions known from the equilibrium theory:

$$G^R(1,1')_{\sigma,\sigma'} = -i\theta(t_1 - t_{1'})\langle\{\psi_\sigma(1), \psi^\dagger_{\sigma'}(1')\}\rangle, \tag{4}$$

$$G^A(1,1')_{\sigma,\sigma'} = +i\theta(t_{1'} - t_1)\langle\{\psi_\sigma(1), \psi^\dagger_{\sigma'}(1')\}\rangle, \tag{5}$$

where $\psi_\sigma$ is the field operator for the electrons with spin $\sigma$ and $\theta$ is the Heaviside function. Retarded and advanced Green's functions provide information about the available states, whereas the off-diagonal element of $\underline{G}$, $G^K$, is the Keldysh Green's function which determines the occupation of the aforementioned states, which is defined as:

$$G^K(1,1')_{\sigma,\sigma'} = -i\langle[\psi_\sigma(1), \psi^\dagger_{\sigma'}(1')]\rangle. \tag{6}$$

We seek to introduce the effect of the nonmagnetic impurity scattering and nuclear spin scattering via a perturbation expansion for the Green's function given in Eq. (3). In nonequilibrium theory, left and right Dyson equations describe the perturbation expansion which utilizes the concept of self energy. The self energy in Keldysh formulation has the same triangular matrix structure as the Green's function given in Eq. (3):

$$\underline{\Sigma}(1,1') = \begin{bmatrix} \Sigma^R(1,1') & \Sigma^K(1,1') \\ 0 & \Sigma^A(1,1') \end{bmatrix}, \tag{7}$$

where the $\Sigma^{R(A)}$ is the retarded (advanced) self energy, whereas $\Sigma^K$ is the Keldysh self energy. Each of these self energy components are also matrices in spin space.

Next, we shall calculate the self energy for each scattering mechanism (nonmagnetic impurity scattering and nuclear spin scattering) and construct the equation of motion by using the left-right subtracted Dyson equation within the gradient approximation:

$$\partial_t \underline{G} + \frac{v_F}{2}\left\{(\hat{z} \times \boldsymbol{\sigma}) \cdot \boldsymbol{\nabla}, \underline{G}\right\} + \frac{i}{\hbar}\left[H_0, \underline{G}\right] = -\frac{i}{\hbar}\left[\underline{\Sigma}, \underline{G}\right], \tag{8}$$

where $H_0$ is the 3DTI Hamiltonian given in Eq. (1).[1] The effects of both nonmagnetic impurity and nuclear spin scattering are manifested via the electron self energy, which we decompose as $\underline{\Sigma} = \underline{\Sigma}_0 + \underline{\Sigma}_m$.

---

[1]Here, we ignore the average Overhauser field generated by finite polarization of the nuclear spins, resulting in the precession of the electron spins.

To derive the transport equation from Eq. (8), we employ the quasiclassical approximation, which relies on the assumption that all the energy scales of the problem is small compared to the Fermi energy $E_F$. The quasiclassical Green's function is defined as:

$$\underline{g}(\boldsymbol{R}, t, \hat{k}, \epsilon) = \frac{i}{\pi} \int d\xi \, \underline{G}(\boldsymbol{R}, t, \boldsymbol{k}, \epsilon), \tag{9}$$

where $\xi = \hbar v_F k - E_F$.

Within the quasiclassical approximation, we obtain the unperturbed retarded and advanced Green's functions for the topological surface states described by Hamiltonian $H_0$ given in Eq. (1):

$$
\begin{aligned}
g^{R/A} &= \frac{i}{\pi} \int d\xi \left( \epsilon - \hbar v_F (\boldsymbol{k} \times \boldsymbol{\sigma}) \cdot \hat{z} + E_F \pm i0^+ \right)^{-1} \\
&\approx \pm \frac{1}{2} \left( 1 + (\hat{k} \times \hat{z}) \cdot \boldsymbol{\sigma} \right),
\end{aligned} \tag{10}
$$

where we regularize the divergent terms in the integral given above by assuming that the Fermi energy scale is the largest energy scale in the problem, i.e. $|\epsilon| \ll \hbar v_F k_F$.

## 2.3 Quantum kinetic equation for the topological surface states

In principle, the equation of motion for the $\underline{G}$ given in Eq. (8) contains the full information about the system. To obtain a kinetic equation we consider the Keldysh component of the equation of motion given in Eq. (8):

$$\partial_t g^K + \frac{v_F}{2} \left\{ (\hat{z} \times \boldsymbol{\sigma}) \cdot \boldsymbol{\nabla}, g^K \right\} + i v_F k_F \left[ (\hat{k} \times \hat{z}) \cdot \boldsymbol{\sigma}, g^K \right] = -\frac{i}{\hbar} \{\Sigma^R, g^K\} + \frac{i}{\hbar} \Sigma^K + \frac{i}{2\hbar} \{(\hat{k} \times \hat{z}) \cdot \boldsymbol{\sigma}, \Sigma^K\}, \tag{11}$$

where we used Eq. (10) and the relation $\Sigma^R = -\Sigma^A$.

The quantum kinetic equation given in Eq.(11) is generic for the surface states of 3D topological insulators without hexagonal warping. The right-hand side of Eq. (11) describe scattering between topological surface states, which consists of two primary contributions: nonmagnetic impurity scattering and nuclear spin scattering. To obtain the collision integrals and the transport equation for the topological surface states, we first focus on the nonmagnetic impurities and calculate their self energy. Later, we shall include the effect of the nuclear spins and derive the transport equations for the overall system.

### 2.3.1 Nonmagnetic impurity scattering

In Fig. 2(a), we show the self energy diagram for the nonmagnetic impurity scattering. We evaluate the self-energy $\underline{\Sigma}_0$ for Gaussian correlated nonmagnetic impurities with zero mean value:

$$\underline{\Sigma}_0(\boldsymbol{R}, t, \epsilon) = n_0 U^2 \int \frac{d^2\mathbf{k}}{(2\pi)^2} \, \underline{G}(\boldsymbol{R}, t, \boldsymbol{k}, \epsilon), \tag{12}$$

where we use the Wigner representation with the center of mass time and position $t \equiv (t_1 + t_2)/2$ and $\boldsymbol{R} \equiv (\boldsymbol{x}_1 + \boldsymbol{x}_2)/2$, respectively. In addition, we take the Fourier transform with respect to the relevant coordinates $\eta = t_1 - t_2$ and $\boldsymbol{r} \equiv \boldsymbol{x}_1 - \boldsymbol{x}_2$ and represent the Green's function in energy($\epsilon$)-momentum($\boldsymbol{k}$) domain. Using Eq. (9), we have:

$$\underline{\Sigma}_0 = -\frac{i}{\tau_0} \langle \underline{g} \rangle, \tag{13}$$

a) $\underline{\Sigma}_0 =$ ━━◀━━

b) $\underline{\Sigma}_m =$ ～◀━

Figure 2: a) The self energy for the nonmagnetic impurity scattering. The dashed line indicates the averaging over the positions of the impurities. b) The self energy for the nuclear spin scattering, where the wiggly line is the nuclear spin correlators. The solid circles in b) represent the nuclear spin scattering vertex, whereas the crosses in a) represent the nonmagnetic impurity scattering vertex. In both a) and b), the solid line represents the electronic Keldysh space Green's function.

where $\langle g \rangle \equiv \int d\hat{k}'/(2\pi) g$ denotes the angular average over the Fermi surface. Here, $\tau_0 = (\pi \nu(E_F) n_0 U^2)^{-1}$ is the nonmagnetic impurity scattering timescale with $\nu(E_F) = E_F/(2\pi\hbar^2 v_F^2)$ being the density of states at the Fermi energy. The elastic mean free path associated with the nonmagnetic impurity scattering is $\ell_{\rm el} = v_F \tau_0$. Using the retarded/advanced quasiclassical Green's function given in Eq. (10), we obtain the retarded/advanced self energy:

$$\Sigma_0^{R/A} = \mp \frac{i}{2\tau_0}. \tag{14}$$

On the other hand, the Keldysh component of the self energy matrix enters the kinetic equation as $\Sigma_0^K = -i\langle g^K \rangle/\tau_0$.

### 2.3.2 Nuclear spin scattering

Next, we focus on the electron self energy arising from the interaction with the nuclear spins. While nuclear spins lack energetic dynamics similar to nonmagnetic impurities, they feature spin dynamics that significantly influence the behavior of electrons. To investigate this further, we consider two-point correlators for the nuclear spins:

$$iD_{\alpha\beta}(1,2) = \left\langle \mathcal{T}_c \left( I_\alpha^{n_1}(t_1) I_\beta^{n_2}(t_2) \right) \right\rangle, \tag{15}$$

where $\alpha, \beta = \{x, y, z\}$ and $\mathcal{T}_c$ denotes the contour ordering [22]. We map the contour ordered nuclear spin correlators onto the Keldysh space and find each element of the nuclear spin correlators as:[2]

$$iD_{\alpha\beta}^R(1,2) = \theta(t_1 - t_2) 2i \delta_{n_1, n_2} \epsilon_{\alpha\beta\gamma} m_\gamma^{n_1}, \tag{16}$$

$$iD_{\alpha\beta}^A(1,2) = -\theta(t_2 - t_1) 2i \delta_{n_1, n_2} \epsilon_{\alpha\beta\gamma} m_\gamma^{n_1}, \tag{17}$$

$$iD_{\alpha\beta}^K(1,2) = 2\delta_{n_1, n_2} \left( \delta_{\alpha\beta} - m_\alpha^{n_1} m_\beta^{n_2} \right), \tag{18}$$

where we define $m_\alpha^n \equiv \langle I_\alpha^n \rangle$ and denote the position of the $i^{\rm th}$ nuclear spin as $n_i$.

Given the nuclear spin correlators, we then calculate the self energy for electrons due to their interaction with the nuclear spins. We show the diagrammatic representation of the self energy due to nuclear spin scattering in Fig. 2(b). We consider two-point correlators for the nuclear spins only and obtain the components of the self energy matrix $\underline{\Sigma}_m$ within the

---

[2]See Appendix B for a detailed derivation of nuclear spin correlators and nuclear spin self energy.

quasiclassical approximation as:

$$\Sigma_m^{R/A}(\boldsymbol{R},\boldsymbol{k},t_1,t_2) = \frac{\lambda^2}{8}\pi\nu(E_F)\int\frac{d\hat{k}'}{2\pi}\Big[\sigma_i\,g^K(\boldsymbol{R},\boldsymbol{k}',t_1,t_2)\,\sigma_j\,D_{ij}^{R/A}(\boldsymbol{R},\boldsymbol{k}-\boldsymbol{k}',t_1,t_2)$$
$$+\,\sigma_i\,g^{R/A}(\boldsymbol{R},\boldsymbol{k}',t_1,t_2)\,\sigma_j\,D_{ij}^K(\boldsymbol{R},\boldsymbol{k}-\boldsymbol{k}',t_1,t_2)\Big],$$

$$\Sigma_m^K(\boldsymbol{R},\boldsymbol{k},t_1,t_2) = \frac{\lambda^2}{8}\pi\nu(E_F)\int\frac{d\hat{k}'}{2\pi}\Big[\sigma_i\,g^K(\boldsymbol{R},\boldsymbol{k}',t_1,t_2)\,\sigma_j\,D_{ij}^K(\boldsymbol{R},\boldsymbol{k}-\boldsymbol{k}',t_1,t_2)$$
$$+\,\sigma_i\,(g^R-g^A)(\boldsymbol{R},\boldsymbol{k}',t_1,t_2)\,\sigma_j\,\big(D_{ij}^R-D_{ij}^A\big)(\boldsymbol{R},\boldsymbol{k}-\boldsymbol{k}',t_1,t_2)\Big],$$

where we use a mixed representation involving $\boldsymbol{R}$ and momentum $\boldsymbol{k}$, as well as the temporal coordinates $t_1$ and $t_2$. Inserting the nuclear spin correlators given in Eq. (16), we obtain

$$\Sigma_m^{R/A} = \frac{\hbar}{\tau_{sf}}\Big[\sigma_i\,\langle g^K\rangle\,\sigma_j\,\big(\pm\theta(\pm(t_1-t_2))\epsilon_{ijk}m_k\big) + \sigma_i\,\langle g^{R/A}\rangle\,\sigma_j\,\big(-i\delta_{ij}\big)\Big],$$
$$\Sigma_m^K = \frac{\hbar}{\tau_{sf}}\Big[\sigma_i\,\langle g^K\rangle\,\sigma_j\,\big(-i\delta_{ij}\big) + \sigma_i\,\big(\langle g^R\rangle - \langle g^A\rangle\big)\,\sigma_j\,\big(\epsilon_{ijk}m_k\big)\Big],$$

where $\tau_{sf}^{-1}\equiv\frac{\lambda^2}{4\hbar}n_m\pi\nu(\epsilon_F)$ is the timescale associated with the mean nuclear spin polarization dynamics, with $n_m$ being the nuclear spin density. Here, we use a coarse grained description of the local mean nuclear spin polarization, namely $m_n\to m(\boldsymbol{R})$. Next, we parameterize the quasiclassical Green's function as $g^K = g_0\sigma_0 + \boldsymbol{g}\cdot\boldsymbol{\sigma}$ and obtain:

$$\Sigma_m^{R/A} = \mp\frac{i\hbar}{\tau_{sf}}\Big[\frac{3}{2}\sigma_0 + \langle\boldsymbol{g}\rangle\cdot\boldsymbol{m}\sigma_0 - \langle g_0\rangle\boldsymbol{m}\cdot\boldsymbol{\sigma}\Big],$$
$$\Sigma_m^K = -\frac{i\hbar}{\tau_{sf}}\Big[3\langle g_0\rangle\sigma_0 - \langle\boldsymbol{g}\rangle\cdot\boldsymbol{\sigma} - 2\boldsymbol{m}\cdot\boldsymbol{\sigma}\Big], \tag{19}$$

where we use $\langle g^{R/A}\rangle = \pm1/2$ and assume that the nuclear spin correlators are independent of momentum and energy. For brevity we drop the arguments of the self energy components, namely the position $\boldsymbol{R}$, time $T$ and energy $\epsilon$.[3]

Having obtained the nonmagnetic impurity-averaged self energy given in Eq. (13) and the self energy due to nuclear spin scattering given in Eq. (19), we can now express the explicit form of the right hand side of Eq. (11). By separating the contributions from nonmagnetic impurity scattering and nuclear spin scattering, each of these contributions can be written as:

$$I_0[g] = -\frac{1}{\tau_0}\Big[g - \langle g\rangle - \frac{1}{2}\{(\hat{k}\times\hat{z})\cdot\sigma,\langle g\rangle\}\Big], \tag{20}$$

and

$$I_m[g] = -\frac{3}{\tau_{sf}}\Big[g - \langle g_0\rangle\sigma_0 + \frac{2}{3}\langle\boldsymbol{g}\rangle\cdot\boldsymbol{m}g - \frac{2}{3}\langle g_0\rangle\boldsymbol{m}\cdot\boldsymbol{\sigma}g_0 - \frac{2}{3}\langle g_0\rangle\boldsymbol{m}\cdot\boldsymbol{g}\sigma_0 + \frac{1}{3}\langle\boldsymbol{g}\rangle\cdot\boldsymbol{\sigma}$$
$$+\frac{2}{3}\boldsymbol{m}\cdot\boldsymbol{\sigma} - \langle g_0\rangle(\hat{k}\times\hat{z})\cdot\boldsymbol{\sigma} + \frac{1}{3}(\hat{k}\times\hat{z})\cdot\langle\boldsymbol{g}\rangle\sigma_0 + \frac{2}{3}\boldsymbol{m}\cdot(\hat{k}\times\hat{z})\sigma_0\Big]. \tag{21}$$

We proceed by inserting Eq. (20) and Eq. (21) into Eq. (11). This way, we obtain the quantum kinetic equation, which we use to derive the transport equations for the surface states in the next subsection.

---

[3]$\Sigma^{R/A}$ contain terms that arise from the Fourier transformation with respect to the relative time coordinate $\eta = t_1-t_2$, which describe the nuclear spin mediated electron-electron interaction. However, as their contribution is not significant compared to the electron dynamics, we ignore these terms.

## 2.4 Transport equations for the topological surface states

Upon obtaining the quantum kinetic equation, we perform a moment expansion to derive the transport equations for the topological surface states. The moment expansion allows us to obtain a hierarchy of equations, which we truncate at a certain order to extract the desired transport equations. We start by taking the trace of Eq. (11) and obtain:

$$\partial_t g_0 + v_F \hat{k} \cdot \boldsymbol{\nabla} g_0 = -\frac{1}{\tau_0}\left[ g_0 - \langle g_0 \rangle + \left(\hat{k} \times \langle \boldsymbol{g} \rangle\right)_z \right]$$
$$-\frac{3}{\tau_{sf}}\left[ g_0 - \langle g_0 \rangle - \frac{1}{3}\left(\hat{k} \times \langle \boldsymbol{g} \rangle\right)_z + \frac{2}{3}\boldsymbol{m} \cdot \left(g_0 \langle \boldsymbol{g} \rangle - \langle g_0 \rangle \boldsymbol{g} + (\hat{k} \times \hat{z})\right) \right]. \quad (22)$$

As $E_F$ is the largest energy scale in the problem, the dominant contribution of the nonequilibrium state $g$ is diagonal in the eigenstates of $H_0$ [5]. In first order, we find the following spin dependent contributions to the Green's function

$$g_x = \hat{k}_y g_0,$$
$$g_y = -\hat{k}_x g_0. \quad (23)$$

Assuming nonmagnetic impurity scattering to be the dominant source of scattering and neglecting the nuclear spin contribution initially, we use Eq. (23) to obtain the subdominant term $g_z$ as:

$$g_z \approx \frac{1}{2v_F k_F}\left( \frac{\hat{k}_x}{\tau_0}\langle \hat{k}_y g_0 \rangle - \frac{\hat{k}_y}{\tau_0}\langle \hat{k}_x g_0 \rangle + v_F \hat{k}_y \nabla_x g_0 - v_F \hat{k}_x \nabla_y g_0 \right), \quad (24)$$

where we see that the term $g_z$ is only nonzero for the first order in $(k_F \ell_{\text{el}})^{-1}$. Notably, we consider a scenario where spin transport is not diffusive. Therefore, we neglect the first order corrections to Eq. (23). Subsequently, we insert this set of equations back into Eq. (22), yielding

$$\partial_t g_0 + v_F \boldsymbol{\nabla} \cdot \hat{k} g_0 = -\frac{1}{\tau_0}\left[ g_0 - \langle g_0 \rangle - \hat{k} \cdot \langle \hat{k}' g_0 \rangle \right]$$
$$-\frac{3}{\tau_{sf}}\left[ g_0 - \langle g_0 \rangle + \frac{1}{3}\hat{k} \cdot \langle \hat{k}' g_0 \rangle + \frac{2}{3}\boldsymbol{m} \cdot \left(g_0 \langle g_0(\hat{k}' \times \hat{z}) \rangle - \langle g_0 \rangle g_0(\hat{k} \times \hat{z}) + (\hat{k} \times \hat{z})\right) \right]. \quad (25)$$

This equation is the quantum kinetic equation for the charge sector of the topological surface states, interacting with both nonmagnetic impurities and nuclear spins. Upon performing an angular average over the quantum kinetic equation for the charge sector we obtain the continuity equation:

$$\partial_t n + 2v_F \left(\boldsymbol{\nabla} \times \boldsymbol{s}\right) \cdot \hat{z} = 0. \quad (26)$$

Correspondingly, we obtain the equation for the energy-resolved spin density using the quantum kinetic equations for the spin sector (see Appendix A for the details of the calculation):

$$\partial_t s_x + \frac{v_F}{4}\nabla_y n + \frac{s_x}{2\tau_0} = \Gamma_x,$$
$$\partial_t s_y - \frac{v_F}{4}\nabla_x n + \frac{s_y}{2\tau_0} = \Gamma_y. \quad (27)$$

We observe that Eq. (26) and Eq. (27) form a set of spin-charge coupled transport equations. In the above equations, we use the generalized density matrix $F(\epsilon, \boldsymbol{R}) = n(\epsilon, \boldsymbol{R})/2\sigma_0 + \boldsymbol{s}(\epsilon, \boldsymbol{R}) \cdot \boldsymbol{\sigma}$,

associated with the angular average of the quasiclassical Keldysh Green's function. Here, we define $\Gamma_i$ as the nuclear spin contribution to the transport equations:

$$\Gamma_x = -\frac{1}{\tau_{sf}}\left[s_x - m_x\left(\frac{n}{2}\left(1-\frac{n}{2}\right)+s_x^2\right)\right],\tag{28}$$

$$\Gamma_y = -\frac{1}{\tau_{sf}}\left[s_y - m_y\left(\frac{n}{2}\left(1-\frac{n}{2}\right)+s_y^2\right)\right],\tag{29}$$

where we redefine the timescale $4\tau_{sf} \equiv \tau_{sf}$. We note that in the absence of nuclear spin scattering, the terms $\Gamma_i$ vanish.[4]

We examine transport dominated by nonmagnetic impurity scattering, specifically when $\tau_0 \ll \tau_{sf}$. Under this condition, transport becomes diffusive, and we solve the corresponding equations in the quasi-stationary state ($\omega\tau \ll 1$). At the lowest order, the energy-resolved spin density becomes

$$s_{x(y)} = \mp\frac{\nu_F\tau_0}{2}\nabla_{y(x)}n + 2\tau_0\Gamma_{x(y)}.\tag{30}$$

We insert Eq. (30) into the continuity equation given in Eq. (26) and obtain an energy-resolved diffusion equation:

$$\partial_t n - \mathcal{D}\nabla^2 n + 4\ell_{\mathrm{el}}(\boldsymbol{\nabla}\times\boldsymbol{\Gamma})\cdot\hat{z} = 0,\tag{31}$$

where we define $\mathcal{D} = \nu_F^2\tau_0$ as the diffusion constant [5]. We identify the energy resolved particle current density as $\boldsymbol{j}(\epsilon,\boldsymbol{R}) = -\mathcal{D}\nabla n + 4\ell_{\mathrm{el}}(\boldsymbol{\Gamma}\times\hat{z})$. Complementary to the electron dynamics, we next obtain the nuclear spin dynamics in the next subsection. By examining the nuclear spin dynamics and identifying the term $\boldsymbol{\Gamma}$, we gain insights into the interplay between electron and nuclear spin interactions.

## 2.5 Nuclear spin polarization dynamics

We now focus on the dynamics of the nuclear spin polarization under the influence of nonequilibrium electron spin polarization and establish its connection to the source term $\boldsymbol{\Gamma}$ in Eq. (31). In order to describe the dynamics of the nuclear spin polarization, we first write down the kinetic equation of nuclear spins. To simplify our analysis, we find it more convenient to work with the lesser/greater Green's function rather than the Keldysh formalism. In the Wigner representation, the kinetic equation for the lesser component of the nuclear spin correlators takes the following form:

$$\dot{D}_{\alpha\beta}^{-+}(\boldsymbol{q},\Omega) = -\frac{i}{\hbar}\left[\Pi_{\alpha\delta}^{-+}(\boldsymbol{q},\Omega)D_{\delta\beta}^{+-}(\boldsymbol{q},\Omega) - D_{\alpha\delta}^{-+}(\boldsymbol{q},\Omega)\Pi_{\delta\beta}^{+-}(\boldsymbol{q},\Omega)\right],\tag{32}$$

where $D$ is the nuclear spin correlators, $\Pi$ is the nuclear spin self energy due to interactions with topological surface states, $\boldsymbol{q}$ is the momentum and $\Omega$ is the frequency. Here, we omitted the position and time variables for clarity.

The above equation of motion for nuclear spin polarization provides insights into the interaction between nuclear spins and nonequilibrium electron spin polarization. We begin our analysis by considering the nuclear spin self energy $\Pi$:

$$\Pi_{\alpha\beta}^{-+}(1,2) = -i\frac{\lambda^2}{4}\operatorname{Tr}\left[\sigma_\alpha G^{-+}(1,2)\sigma_\beta G^{+-}(2,1)\right],\tag{33}$$

---

[4]In this case, the only scattering is due to nonmagnetic impurities, and we recover the results obtained by Ref. [5].

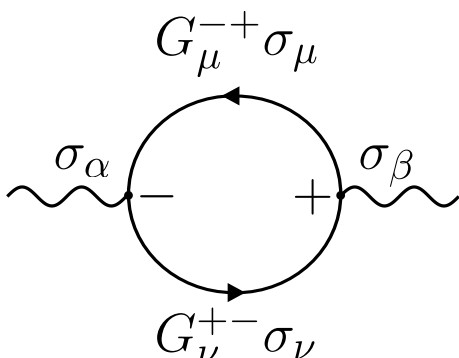

Figure 3: The diagrammatic representation for the lesser component of the nuclear spin self energy $\Pi_{\alpha\beta}^{-+}$.

where the trace is over the spin degree of freedom. Here, we use the lesser (greater) component of the electronic Green's function, namely $G^{-+}(G^{+-})$, for convenience.[5]

In Fig. 3, we present the diagrammatic representation of the lesser component of the nuclear spin self energy. To calculate the nuclear spin self-energy $\Pi^{-+}$, we follow a similar approach as with the electron self-energy. Using the Wigner representation for $\Pi^{-+}$ and then taking the Fourier transform with respect to the relative coordinates, we obtain:

$$\Pi_{\alpha\beta}^{-+}(\boldsymbol{q},\Omega) = -i\frac{\lambda^2}{4} \int \frac{d^2\boldsymbol{k}}{(2\pi)^2} \int \frac{d\omega}{2\pi} \, \mathrm{Tr}\left[\sigma_\alpha G^{-+}(\boldsymbol{k},\omega)\sigma_\beta G^{+-}(\boldsymbol{k}-\boldsymbol{q},\omega-\Omega)\right]. \tag{34}$$

We proceed by parametrizing the electronic Green's function, namely $G \equiv G_\mu \sigma_\mu$ with $\mu = \{0, x, y, z\}$ and calculate the lesser and greater components of the nuclear spin self energy (see Appendix C).

Using the nuclear spin self energy and nuclear spin correlators, we formulate the quantum kinetic equation for the lesser components of the momentum integrated nuclear spin correlator, $d_{\alpha\beta}^{-+}$ (see Appendix C):

$$\dot{d}_{\alpha\beta}^{-+}(\boldsymbol{r},t) = -\frac{i}{\hbar}\left(\pi_{\alpha\delta}^{-+} d_{\delta\beta}^{+-}(\boldsymbol{r},t) - \pi_{\alpha\delta}^{+-} d_{\delta\beta}^{-+}(\boldsymbol{r},t)\right), \tag{35}$$

where the term on the right hand side describes the spin-flip interaction taking place between nuclear spins and electron spins. Here, $\pi^{\mp\pm}$ describes the nuclear spin self energy components, integrated over the momentum $\boldsymbol{q}$. Inserting the nuclear spin self energy into Eq. (35), we obtain the equation for the nuclear spin polarization dynamics:

$$\dot{m}_\gamma(\boldsymbol{r},t) = -\frac{\lambda^2 \epsilon_F^2}{4\pi(\hbar v_F)^4} \int \frac{d\epsilon}{\hbar} \, m_\gamma(\boldsymbol{r})\left(\frac{n(\epsilon,\boldsymbol{r})}{2}\left(1-\frac{n(\epsilon,\boldsymbol{r})}{2}\right)+s_\gamma^2(\epsilon,\boldsymbol{r})\right)-s_\gamma(\epsilon,\boldsymbol{r}), \tag{36}$$

where we use the relation $d_{\alpha\beta}^{-+} = \epsilon_{\alpha\beta\gamma} m_\gamma(\boldsymbol{r})$ (see Appendix B) for the case $\alpha \neq \beta$ with $\gamma \in \{x, y\}$. Here, we consider a coarse grained description and define the average nuclear spin polarization $\boldsymbol{m}(\boldsymbol{r})$. The equations describing the dynamics of nuclear spin polarization in Eq. (36) are general for the Fermi contact interaction. However, the density of states and the electron spin density depend on the electronic part of the Hamiltonian, which is influenced by the topological surface states of the 3DTI. Incorporating the effect of the surface states via the electron density matrix, we establish a direct connection between the nuclear spin dynamics

---

[5]We note that the Keldysh representation of the electronic Green's function is related to the representation used here via a linear transformation [22].

and the source term $\boldsymbol{\Gamma}$ in the diffusion equation presented in Eq.(31). We identify the integrand in the right-hand side of Eq.(36) as the source terms $\Gamma_\gamma$ in Eq. (28). We then express the nuclear spin dynamics in a more concise form as:

$$\frac{d\mathbf{m}}{dt} = -\nu \int d\epsilon\,\boldsymbol{\Gamma}\,, \tag{37}$$

where $\nu$ is the density of states of the electron subspace. Here, we define $\mathbf{m}(\boldsymbol{r}) \equiv n_m \boldsymbol{m}(\boldsymbol{r})/2$ as the nuclear spin polarization density. We note that the energy integral of the source term $\boldsymbol{\Gamma}$ is related to the time rate of change of mean nuclear spin polarization density $\mathbf{m}$. We emphasize that Eq. (37) offers a generic description of nuclear spin dynamics interacting with electron spins through the Fermi contact interaction, while the specific form of the term $\boldsymbol{\Gamma}$ is determined by the characteristics of the system under consideration. In the following subsection, we apply the generalized nuclear spin dynamics from Eq. (37) to the topological surface states of a 3DTI. By doing so, we obtain a formula for the nuclear spin polarization and its connection to the induced charge current.

## 2.6 Entropy-induced charge current

Integrating Eq.(31) over energy and utilizing Eq.(37), we derive the diffusion equation for the charge density:

$$\partial_t \rho - D\nabla^2 \rho + 4e\ell_{\text{el}}\boldsymbol{\nabla}\cdot\left(\frac{d\mathbf{m}}{dt}\times\hat{z}\right) = 0\,. \tag{38}$$

Here, $\rho$ represents the charge density, defined as $\rho \equiv -e\nu/2 \int d\epsilon\, n + \nu e^2 \phi$, where $\phi$ denotes the scalar electrostatic field and $e$ is the elementary charge. The charge current density arising from the diffusion equation in Eq. (38) is given by:

$$\boldsymbol{J}(\boldsymbol{r},t) = -D\boldsymbol{\nabla}\rho + 4e\ell_{\text{el}}\left(\frac{d\mathbf{m}}{dt}\times\hat{z}\right)\,. \tag{39}$$

In this expression, we observe that the time rate of change of the nuclear spin polarization density $\mathbf{m}$ acts as a charge current source. This is a result of the spin-momentum locking feature of the surface states: as nuclear spin polarization is transferred to electron spins, the perfect spin-momentum locking induces a charge current.

We now investigate the effect of the nuclear spin polarization dynamics in a setup depicted in Fig. 1. The setup consists of a 3D topological insulator with two reservoirs connected to its top surface. We focus only on the top surface and assume negligible hybridization between the top and bottom surface states. Moreover, for the sake of demonstration, we consider the nuclear spin polarization density $\mathbf{m}$ to have a weak position dependence, allowing us to consider its position-independent contribution for simplicity.

We start by considering a setup where a voltage bias is applied between the reservoirs, leading to a charge current flowing along the $x$-direction. In this case, the transport is described by a one-dimensional diffusion equation. Solving this equation allows us to determine the charge current, which is given by:

$$I = GV - 4eN\frac{\ell_{\text{el}}}{L}\frac{dm_y}{dt}\,, \tag{40}$$

where $V$ is the applied voltage bias. Here, we use the relation $I = JW$ with $W$ being the width of the surface along $y-$direction. The first term on the right-hand side corresponds to the familiar Ohm's law with the conductance given by $G = \sigma W/L$, where $\sigma = e^2 \nu D$ represents

the conductivity obtained from Einstein's relation (not to be confused with Pauli matrices in spin space). This bare conductance $G$ solely depends on the nonmagnetic impurity scattering. The second term is the nuclear spin dynamics induced charge current with $N$ being the total number of nuclear spins at the top surface of a 3DTI and $m_y$ being the average nuclear spin polarization in the $y-$direction.

To investigate the relationship between the time rate of change of nuclear spin polarization and the applied voltage, we solve the nuclear spin dynamics governed by Eq. (37) under an applied voltage bias:

$$\frac{dm_y}{dt} = \frac{\gamma_0^{3D}}{\hbar} \left\{ \frac{\ell_{\text{el}}}{L} \frac{eV}{2} - m_y \left[ \frac{\ell_{\text{el}}}{L} eV \coth \left( \frac{\ell_{\text{el}}}{L} \frac{eV}{2k_B T} \right) \right] \right\}, \tag{41}$$

where $T$ is the temperature of the reservoirs and $\gamma_0^{3D}$ is the effective interaction strength between nuclear spins and topological surface states:

$$\gamma_0^{3D} \equiv \frac{\lambda^2 v^2}{4} = \frac{1}{8\pi} \frac{v_0^2}{\xi^2} \left( \frac{E_F}{\hbar v_F} \right)^2 \left( \frac{A_0}{\hbar v_F} \right)^2. \tag{42}$$

By inserting Eq. (41) into Eq. (40), we determine the current-voltage characteristics of a 3DTI in the presence of a nuclear spin bath:

$$I = GV - 4eN \frac{\ell_{\text{el}}}{L} \frac{\gamma_0^{3D}}{\hbar} \left\{ \frac{\ell_{\text{el}}}{L} \frac{eV}{2} - m_y \left[ \frac{\ell_{\text{el}}}{L} eV \coth \left( \frac{\ell_{\text{el}}}{L} \frac{eV}{2k_B T} \right) \right] \right\}. \tag{43}$$

This equation is one of the central results of this work. Notably, the first term in the parenthesis on the right hand side of Eq. (43) is a dissipative term due to nuclear spin scattering and simply renormalizes the bare conductance $G$. This dissipative term vanishes in the absence of an applied voltage bias. On the other hand, the second term in the parenthesis is non-vanishing even in the absence of an applied voltage bias, as long as there exists a finite nuclear spin polarization $m_y$. Remarkably, finite nuclear spin polarization $m_y$ induces a charge current, converting the thermal energy in the environment into electrical energy. This induced current reads

$$I_{\text{ind}} = 4eN \frac{\ell_{\text{el}}}{L} \frac{m_y}{\tau_m}, \tag{44}$$

where we define the characteristic time scale $\tau_m = \hbar/(2k_B T \gamma_0^{3D})$.

Examining this from an entropy point of view provides valuable insight: Finite nuclear spin polarization has a lower entropy, driving nuclear spins towards a state of higher entropy. This transition can only be achieved by transferring their spin angular momentum to electron spins via hyperfine interaction. Consequently, entropy is transferred from the reservoirs to the nuclear spins, resulting in an entropy-induced charge current.

Furthermore, the magnitude of the induced current presented in Eq.(44) incorporates a scaling factor $\ell_{\text{el}}/L$, indicating that the presence of nonmagnetic impurity scattering counteracts the induced charge current. We emphasize that this is a feature of the topological surface states of a 3DTI. The randomization of the spin of charge carriers due to nonmagnetic impurity scattering leads to a reduction in the magnitude of the induced charge current. This is in contrast with the helical edge states of a quantum spin Hall insulator, where nonmagnetic impurity scattering solely leads to forward scattering without altering the spin of the charge carriers. Consequently, the scaling factor observed in Eq. (44) is not present in quantum spin Hall insulators [1].

# 3  Nuclear spin-driven inductance of topological insulators

The current-voltage characteristic for the topological surface states (TSS) of a 3DTI given in Eq. (43) includes an additional contribution to the current, set by the nuclear spin (de)polarization rate. In this section, we define an electronic circuit element with an equivalent response, showing that this additional contribution has an inductive nature.

For simplicity, we again assume that nuclear spins are polarized in $y$-direction only, $m_y = m$. Moreover, we consider the limit $eV\ell_{\rm el}/L \ll 2k_B T$, which applies when the voltage drop over a mean free path length is small compared to the thermal energy. The subsequent analysis considers probing the circuit using a sinusoidal signal at frequency $\omega = \omega_0$. Albeit not the most general case, this limit still provides an intuitive picture of the current-voltage characteristics.

## 3.1  Equivalent circuit configuration

In the limit $eV\ell_{\rm el}/L \ll 2k_B T$, Eq. (41) reduces to

$$\frac{{\rm d}m}{{\rm d}t} = \frac{\gamma_0^{\rm 3D}}{\hbar}\left[\frac{\ell_{\rm el}}{L}\frac{eV}{2} - 2mk_B T\right]. \tag{45}$$

We investigate the impedance of the topological surface states at frequency $\omega_0$ using a time-dependent applied voltage bias $V(\omega_0, t) = V_0 e^{i\omega_0 t}$ and assuming $m(\omega_0, t) = |m|e^{i\omega_0 t + \phi}$. In this case, Eq. (43) reduces to

$$I(\omega_0) = \left[G - 4\pi e^2 N \frac{\gamma_0^{\rm 3D}}{h}\left(\frac{\ell_{\rm el}}{L}\right)^2 \frac{i\omega_0 \tau_m}{i\omega_0 \tau_m + 1}\right]V_0, \tag{46}$$

where $\tau_m = \frac{\hbar}{2k_B T \gamma_0^{\rm 3D}}$ is the characteristic nuclear polarization timescale. Eq. (46) consists of two terms: an Ohmic contribution, and a non-Ohmic (nuclear-polarization induced) contribution to the current-voltage characteristics. Next, we investigate how the Ohmic contribution and nuclear polarization-induced contribution scale relative to each other.

We consider a device with a surface of width $W$ and length $L$. The (bare) Ohmic contribution is given by the Drude conductivity as

$$I_{\rm Ohmic} = GV = G_0 \frac{k_F l_{el}}{\pi}\frac{W}{L}V. \tag{47}$$

The nuclear spin polarization-induced current depends on the device geometry through the ratio $\ell_{\rm el}/L$ and through $N = [N]WL\xi/v_0$ with $[N]$ being the number of nuclear spins per unit cell. Taking both contributions into account, we define the impedance response of a topological surface state as:

$$I(\omega_0) = G\left[1 - \zeta^{\rm 3D}\frac{i\omega_0 \tau_m}{i\omega_0 \tau_m + 1}\right]|V| \equiv Z_{\rm TSS}^{-1}(\omega_0)V_0, \tag{48}$$

where

$$\zeta^{\rm 3D} = 4\pi^2[N]\gamma_0^{\rm 3D}\frac{\xi\ell_{\rm el}}{k_F v_0}, \tag{49}$$

is a dimensionless parameter that captures the relative magnitude of the induced current with respect to the Ohmic contribution. We show the frequency dependence of $Z_{\rm TSS}$ in Fig. 4(b). Note that as $\omega_0 \to 0$, the induced current vanishes, i.e. $I_{\rm ind} \to 0$. On the other hand, as $\omega_0 \to \infty$, we have $I_{\rm ind}$ real and maximum. Furthermore, we point out that the out-of-phase component of $I_{\rm ind}$ is maximum for $\omega_0 = \tau_m^{-1}$.

We find an impedance exhibiting a frequency response analogous to Eq. (48) by using a combination of an inductor and resistor parallel to a second resistor. We illustrate this circuit configuration in Fig. 4(a), with its corresponding current response given as:

$$
\begin{aligned}
I_{\text{LRR}}(\omega_0) &= \left[ \frac{1}{i\omega_0 \mathcal{L} + R_{\text{series}}} + \frac{1}{R_{\text{shunt}}} \right] V_0 \\
&\equiv I_{\mathcal{L}} + I_{\text{shunt}}.
\end{aligned}
\tag{50}
$$

This is equivalent to the TSS under the conditions

$$
\begin{aligned}
(R_{\text{series}})^{-1} &= G\zeta^{3D}, \\
(R_{\text{shunt}})^{-1} &= G\left(1 - \zeta^{3D}\right), \\
\mathcal{L} &= \frac{\tau_m}{G\zeta^{3D}}.
\end{aligned}
\tag{51}
$$

The equivalent circuit model forms the key result of this section: the presence of nuclear spins interacting with the topological surface states through hyperfine interaction leads to an inductive response in the current-voltage relation.

Fig. 4(c) shows the resistance values in the equivalent circuit model as a function of the hyperfine coupling strength $\zeta^{3D}$. For vanishing hyperfine coupling strength ($\zeta^{3D} = 0$), the charge current through the inductive branch of the circuit element ($I_{\mathcal{L}}$) vanishes, reducing Eq. (48) to Ohm's law describing diffusive transport. Note that $I_{\mathcal{L}}$ is different from $I_{\text{ind}}$ defined previously. $I_{\text{ind}}$ is the total contribution of hyperfine coupling to the current, which *subtracts* from the $\omega_0 \ll \tau_m^{-1}$ value (given by $I_{\text{Ohmic}}$). On the other hand, in the equivalent circuit the current in *both* branches is influenced by hyperfine coupling (51), and $I_{\mathcal{L}}$ *adds* to the $\omega_0 \gg \tau_m^{-1}$ value (given by $R_{\text{shunt}}^{-1}V = I_{\text{Ohmic}} - G\zeta^{3D}V$).

We characterize the efficiency of the inductive effect by evaluating the quality factor at the operating frequency $\omega_0 = \tau_m^{-1}$ in Fig. 4(b), as

$$
Q \equiv \left| \frac{Im(Z_{\text{TSS}})}{Re(Z_{\text{TSS}})} \right| = \frac{R_{\text{shunt}}}{2R_{\text{series}} + R_{\text{shunt}}} = \frac{\zeta^{3D}}{2 - \zeta^{3D}}.
\tag{52}
$$

We note that while the quality factor is enhanced as $\zeta$ increases, the model described in this section breaks down for $\zeta > 1$. Considering Eq. (42) and Eq. (49), this implies that a large $k_F$ is favorable for observing an induced current due to finite nuclear spin polarization. However, the maximally attainable $k_F$ is limited by the bulk band gap, as bulk carriers would introduce additional Ohmic shunt channels, thereby reducing $Q$.

Apart from analyzing the efficiency at the operating frequency $\omega = \tau_m^{-1}$, we can investigate the inductive effect by considering the low-frequency (quasi-DC) limit of equation 48. For $\omega\tau_m \ll 1$, this reduces to a relation equivalent to a series combination of one resistor and inductor:

$$
V(\omega_0) \approx I(\omega_0)\left(G^{-1} + i\omega_0 \mathcal{L}_{\omega\tau_m \ll 1}\right),
\tag{53}
$$

where we define

$$
\mathcal{L}_{\omega\tau_m \ll 1} = G^{-1}\zeta^{3D}\tau_m = 4\pi^3 \frac{N}{n_{\text{modes}}^2} \frac{1}{G_0} \frac{\hbar}{2k_B T},
\tag{54}
$$

where $n_{\text{modes}} = k_F W$ is the number of modes contributing to conduction along the width of the device. This low-frequency inductance value is universal for different materials, as it is independent of hyperfine coupling energy (although it depends on the number of nuclear spins, and will reduce to zero when the material has zero nuclear spin). The other parameters can be tuned by varying the Fermi energy, geometry and temperature. Filling in the parameters at $T = 100$ K gives $\mathcal{L}_{\omega\tau_m \ll 1} = (N/n_{\text{modes}}^2) \cdot 122$ nH. The remarkable feature of the universal

TSS inductance is that it depends on the aspect ratio ($N \sim LW$, $n_{\text{modes}} \sim W$), and not on the absolute area of the device as in conventional (magnetic) inductors. Therefore, topological surface states offer a means of miniaturization of inductive circuit elements. However, for application purposes, the relative magnitude of the inductive effect with respect to the Ohmic shunt current is the relevant quantity which will be further touched upon in section 3.2.

To conclude this section, we need to address two points. Firstly, we emphasize that when we analyze the device in the low bias limit for the complete frequency range, we can describe the topological surface states interacting with nuclear spins as a circuit element with an inductive component. However, taking this limit inherently limits the quality factor. The mean nuclear spin polarization achieved during each polarization cycle is exceedingly small, resulting in a suppression of the induced current $I_{\text{ind}}$ proportional to the rate of change of nuclear spin polarization $dm/dt$. To resolve this, we can enhance $I_{\text{ind}}$ by polarizing (charging) the nuclear spins coupled to the TSS in the limit $eV \gg 2k_B T L/\ell_{\text{el}}$, and depolarizing (discharging) when $eV \ll 2k_B T L/\ell_{\text{el}}$. This way, we ensure a higher initial polarization during the discharging phase, while limiting the Ohmic contribution to the current. The inductive nature of the response is maintained, as the finite nuclear polarization obtained during charging conserves the current direction, but the response is no longer limited to a single harmonic. Nevertheless, the lumped element analysis we discussed earlier provides an intuitive perspective on the current-voltage relationship when considering dynamic nuclear polarization coupled to a topological surface state.

Secondly, there is another equivalent circuit configuration with a frequency response identical to $Z_{\text{TSS}}$. This arises from the fact that each three-element circuit has two non-trivial combinations [23]. The second equivalent circuit consists of a parallel resistor/inductor combination in series with a second resistor. Consequently, the presence of an inductive component is independent of the chosen equivalent configuration. Similar to the equivalent circuit discussed above, the resistance and inductance values in the second equivalent circuit cannot be independently adjusted by tuning material parameters. In other words, probing $\mathcal{L}$ without accounting for $R_{\text{series,shunt}}$ is not possible. To summarize, the key takeaway is that a circuit equivalent to the TSS always includes an inductive element.

## 3.2 Efficiency estimate in candidate materials

Analyzing Eq. (48) and Eq. (49), we observe that the magnitude of the induced current with respect to the Ohmic current depends on a dimensionless constant

$$[N]\frac{\xi \ell_{\text{el}}}{k_F v_0} \equiv N_{\text{coh}}, \tag{55}$$

at constant $\gamma_0$ and $T$. Here, we define $N_{\text{coh}}$ as the amount of nuclei per 'coherent segment', or the nuclei that can be polarized coherently within the surface state penetration depth, Fermi wavelength and electronic mean free path. Using this definition and our circuit element analysis, we can generalize the TSS inductive effect to different spin-momentum locked systems. For example, if we consider a quantum spin Hall insulator with helical edge states, the depth and width of a coherent segment correspond to the cross section of this edge state ($S$), and the length equals the device length ($L$). In this limiting case, the inductive effect scales with $\zeta^{\text{QSH}} \sim \gamma_0^{\text{QSH}}[N]SL/v_0$, which corresponds to predictions for quantum spin Hall edge states [1].

Furthermore, describing the inductive response of topological surface states via $N_{\text{coh}}$ and $\gamma_0$ enables us to estimate material suitability for applications. In Fig. 5, we compare the ratio of induced current to Ohmic current for a selection of materials with topological surface states of 3DTIs or helical edge states of quantum spin Hall insulators. Varying $k_F$ by applying a top or bottom gate potential, we change $N_{\text{coh}}$ for topological surface states. Increasing $k_F$

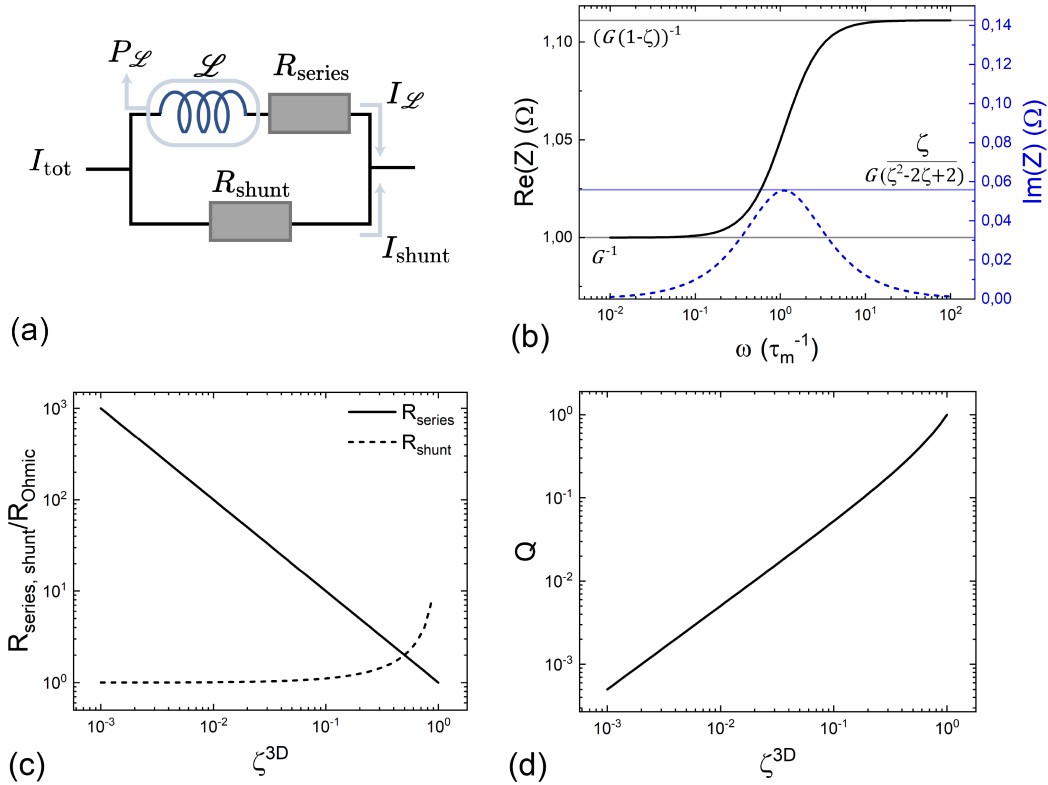

Figure 4: Equivalent circuit model of the topological surface states, represented by two resistors and an inductor in parallel. (a) The electronic circuit configuration, (b) the frequency response of the real and imaginary part of $Z_{\text{TSS}}$, (c) $R_{\text{series}}$ (solid), $R_{\text{shunt}}$ (dashed) relative to $R_{\text{Ohmic}} \equiv G^{-1}$ and (d) the quality factor as a function of $\zeta^{\text{3D}}$.

reduces $N_{\text{coh}}$ by decreasing the Fermi wavelength while enhancing $\gamma_0^{\text{3D}}$. For helical edge states of quantum spin Hall insulators, the density of states does not depend on the Fermi wavevector $k_F$. Thus, we vary $N_{\text{coh}}$ through the device length $L$, not influencing $\gamma_0^{\text{QSH}}$ [1].

In Fig. 5, we show our estimations for device lengths ranging from 10 nm to 10 $\mu$m, but the exactly attainable value may vary in an experiment. Our first example is the alloy $(\text{Bi}_{1-x}\text{Sb}_x)_2\text{Te}_3$ (BST), an experimentally available 3DTI. The position of the Dirac point and Fermi level can be tailored by adjusting the stoichiometry [7]. Furthermore, by reducing the film thickness of BST, the surface states hybridize, opening a gap at the Dirac point. With the appropriate choice of film thickness, this gap becomes nontrivial [24], transitioning the material into the quantum spin Hall state. Bismuth has 9/2 nuclear spin at 100% abundance, whereas antimony has 7/2 nuclear spin at 57% abundance and 5/2 at 43% abundance [25]. To estimate the hyperfine coupling strength, we take the value for bismuth at 50 $\mu$eV, although this will be reduced due to the p-like nature of the surface states [26], similar to hyperfine coupling strength for HgTe-based topological insulators [27]. Therefore, we estimate the total hyperfine coupling strength of BST to be within $5-50\,\mu$eV, consistent with the values reported in the literature [28,29]. Typical values are $\ell_{\text{el}} \sim 10$ nm, $v_F \sim 3-5 \cdot 10^5$ m/s, a bulk band gap of typically 300 $\mu$eV (as an upper limit to gate-tunability of the Fermi level) [7,30,31], and we estimate the penetration depth $\xi \sim 1$nm. For the corresponding quantum spin Hall state, we assume a film thickness of 3 nm, with a decay length of $\sim 10$ nm [30].

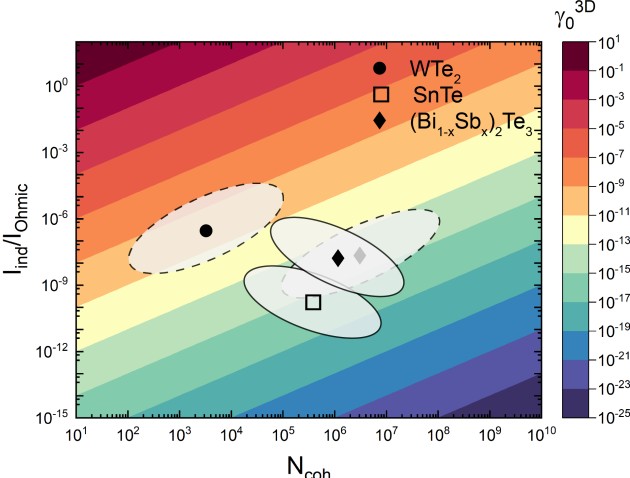

Figure 5: Relative induced current ($\gamma_0 N_{\text{coh}}$), estimated for different materials hosting spin-momentum locked states. Background colors correspond to varying $\gamma_0$ values. Solid outlines denote the surface states of 3DTIs, where $N_{\text{coh}}$ is varied through $k_F$. Dashed outlines denote quantum spin Hall insulators, where $N_{\text{coh}}$ is varied through channel length.

Our second example is the topological insulator SnTe. In this material only the Te atoms have a nonzero nuclear spin. Therefore, we only need to consider the Fermi contact interaction of Te, which is equal to $A_0 = 8.3$ $\mu$eV [32]. Moreover, we use typical values of $\ell_{\text{el}} \sim 200$ nm [33], $\xi \sim 2$ nm [34], $v_F \sim 6 \cdot 10^5$ m/s [35], and limit $E_F$ to the band gap of 180meV [36]. Although $N_{\text{coh}}$ is similar between BST and SnTe, the weaker hyperfine interaction diminishes the efficiency of the induced current in the latter.

The last example considered in our estimate is monolayer WTe$_2$, a quantum spin Hall insulator. On average, the hyperfine coupling constant falls within the range of 0.6 - 6 $\mu$eV, depending on whether the interaction primarily arises from tungsten atoms or is averaged between tungsten and tellurium [32]. The lattice constants are $a = 6.33$, $b = 3.47$, $c = 14.07$ [37], and $v_F \sim 10^5$ m/s [38]. These parameters result in a strong hyperfine coupling compared to the quantum spin Hall states in BST. However, WTe$_2$ has the drawback of a low density of nuclear spins, with an average of 10% being spinful. Additionally, the helical edge states in WTe$_2$ decay over a length of approximately 1.8nm [38]. Low nuclear spin density and reduced cross section in WTe$_2$ diminishes $N_{\text{coh}}$ and consequently $I_{\text{ind}}/I_{\text{Ohmic}}$, counteracting the benefits of increased coupling strength.

Finally, to potentially increase the induced current, we can explore Fermi arc surface states [39]. These states also feature spin-momentum locking and do not suffer from backscattering. Moreover, they extend across the entire surface by forming a coherent quantum state, thus, all nuclear spins coupled to these states contribute to the induced current, without reducing the efficiency due to impurity scattering in topological surface states. By adjusting the width of the device, we can control the number of nuclei per coherent segment, effectively running multiple quantum spin Hall edge states in parallel. If the hyperfine interaction strength is similar to the quantum spin Hall insulator case, $I_{\text{ind}}/I_{\text{Ohmic}}$ can increase by multiple orders of magnitude with respect to the quantum spin Hall state. Therefore, a promising avenue for future research lies in investigating surface states and Fermi arcs of Weyl semi-metals [40,41].

# 4 Entropic nature of inductive response

As demonstrated in the previous section, the dynamic nuclear spin polarization in a spin-momentum locked system gives rise to an inductive response. Unlike a conventional inductor, where energy is stored in a magnetic field generated by current flow, the energy storage mechanism in a 3DTI is distinct. This leads to the question: how is the inductive energy in a 3D topological insulator stored and harnessed? In this section, we reveal that the inductive response is linked to the information entropy contained within the nuclear spin subsystem. Thus, we characterize the topological surface states of a 3DTI as an "entropic inductor".

To establish the link between the inductance of the topological surface states and the entropy of the nuclear spins, we define the information entropy of the nuclear spin subsystem. Assuming a uniform current in the $x$-direction, we account for two spin orientations per nuclear spin: 'up' and 'down', aligned (or anti-aligned) with the average spin polarization of the charging current. This definition is akin to the mean polarization of the nuclear spins, $m = \frac{N_\uparrow - N_\downarrow}{2N}$, given in Ref. [1]. Consequently, the rate of change of entropy of the nuclear spin subsystem is given as

$$\frac{\mathrm{d}S_{\mathrm{nuc}}}{\mathrm{d}t} = \frac{\mathrm{d}}{\mathrm{d}t} \ln \frac{N!}{N_\uparrow! N_\downarrow!} \approx -N \frac{\mathrm{d}m}{\mathrm{d}t} \ln\left(\frac{1+2m}{1-2m}\right). \tag{56}$$

Following Eq. (56), we establish that the induced current and entropy of the nuclear spin subsystem is related: a change in the entropy of the nuclear spin subsystem induces a charge current until equilibrium for the latter ($m = 0$) is reached.

Having found a direct relation between induced current and entropy of the nuclear spin subsystem, we now investigate the relation between the power delivered to the inductor in the equivalent circuit configuration and the entropy of the nuclear spin subsystem. To that end, we calculate the instantaneous power generated by the equivalent circuit configuration of the topological surface states given in Section 3. We note that we consider only the current flowing through the arm containing the inductor $\mathscr{L}$ (and $R_{\mathrm{series}}$). The current flowing through the other arm, including $R_{\mathrm{shunt}}$, merely results in dissipative Joule heating. The instantaneous power delivered to the inductor is

$$P_{\mathscr{L}}(t) = \mathscr{L} I_{\mathscr{L}} \frac{\mathrm{d}I_{\mathscr{L}}}{\mathrm{d}t}. \tag{57}$$

We now assume that the applied voltage bias contains a single harmonic, $V(t) = V_0 e^{i\omega_0 t}$. In this case, we find the complex power delivered to the inductive branch as

$$P_{\mathscr{L}}(\omega_0) = \frac{i\omega_0 \mathscr{L}}{(i\omega_0 \mathscr{L} + R_{\mathrm{series}})^2} V_0^2. \tag{58}$$

Next, we relate the power on the inductive branch to the entropy of the nuclear spin subsystem. In the low-bias frequency regime ($eV \ll 2k_B T L/\ell_{\mathrm{el}}$, as discussed in section 3), the Fourier transformation of Eq. (56) results in

$$i\omega_0 S_{\mathrm{nuc}}(\omega_0) \approx -4N\left(\frac{\gamma_0^{3D} l}{\hbar L}\right)^2 \frac{i\omega_0 \tau_m^2}{(i\omega_0 \tau_m + 1)^2}(eV_0)^2$$
$$= -\frac{1}{k_B T} \frac{i\omega_0 \mathscr{L}}{(i\omega_0 \mathscr{L} + R_{\mathrm{series}})^2} V_0^2. \tag{59}$$

Using Eq. (58) and Eq. (59) and Fourier transforming back to the time domain, we observe that the power delivered by the inductive branch is given by the rate of change of entropy of

the nuclear spin subsystem:

$$P_{\mathscr{L}}(t) = -k_B T \frac{dS_{\text{nuc}}}{dt}. \tag{60}$$

As a change in entropy is linked to heat flowing in/out of the system, this answers the question of where the inductive energy is stored: when polarizing the nuclear spin subsystem, its entropy $S_{\text{nuc}}$ is lowered, and heat is dissipated to the environment. The subsequently induced current upon thermal relaxation is powered by absorbing heat from the environment, until $S_{\text{nuc}}$ reaches its maximum at equilibrium. These considerations underline that the TSS functions as an 'entropic inductor', utilizing the environment to store inductive energy in the form of heat, in contrast to classic inductors utilizing magnetic fields for the same purpose.

Naturally, the induced current must comply with the second law of thermodynamics:

$$P + k_B T \frac{dS_{\text{nuc}}}{dt} \geq 0, \tag{61}$$

where $P$ is the total power, which includes both inductive power $P_{\mathscr{L}}$ and the power dissipated $P_{R_{\text{series,shunt}}}$. These dissipative terms ($P_R = I^2 R > 0$) always result in a positive contribution to the total power of the TSS, ensuring the second law is satisfied.

## 5 Conclusion

In conclusion, we have investigated the topological surface states of a 3D topological insulator interacting with nuclear spins present at the surface through hyperfine interaction. Specifically, we have demonstrated that this interaction leads to dynamical nuclear spin polarization, driven by the Edelstein effect. Furthermore, we have revealed that the reverse process is also possible: a finite nuclear spin polarization induces a charge current response in the system. By using the nonequilibrium Green's function formalism, we have derived the transport equations for the topological surface states of a 3D topological insulator in the presence of nuclear spins and nonmagnetic impurities. Furthermore, we have also obtained the nuclear spin dynamics in the presence of topological surface states and shown that the current-voltage relation of the system is proportional to the rate of change of nuclear spin polarization. This current-voltage relation exhibits an inductive nature in addition to the usual Ohmic response. By modelling this relation as a lumped element model with two parallel resistors and an inductor, we have quantified the quality factor associated with the inductive response of a 3D topological insulator. We have found that the quality factor is enhanced by increasing the coupling parameter $\zeta^{3D}$, possibly reaching unity if $\zeta^{3D}$ is of order one. However, this approximation breaks down for $\zeta^{3D} > 1$.

We have shown that the efficiency of the inductive effect is mainly determined by two factors: the number of nuclear spins within a coherent segment and the strength of the hyperfine interaction between nuclear spins and electrons. These factors remain unaffected by the specific geometry of the device on the surface states of 3D topological insulators. However, we can adjust the efficiency to a certain extent by modifying the Fermi wave number, which can be accomplished by applying a gate voltage. Conversely, for quantum spin Hall insulators, a similar inductive effect is observed but the efficiency does not depend on the Fermi wave number. Instead, efficiency can be adjusted by altering the device's length, as all nuclear spins along an edge fall into a single coherent segment.

In estimating the induced current in various material systems, we focused on the ratio $I_{\text{ind}}/I_{\text{Ohmic}}$ under harmonic excitation. We can apply a similar analysis and find values for the lumped elements constituting the equivalent circuit in Figure 4(a). At low operating frequencies, the analysis results in a universal inductance value of 122 nH, scaled by $N/n_{\text{modes}}^2$.

However, the equivalent inductive circuit element would always be accompanied by $R_{\text{series}}$ and $R_{\text{shunt}}$, and cannot be probed separately. Therefore, a more suitable value to consider for application purposes is the ratio between induced and Ohmic currents. Our analysis shows that this total induced current can be of significant value for the correct choice of material parameters.

Spin accumulation with nuclear spins features inductive properties that could be utilized for technological purposes. This inductive response is a property of spin-momentum locked systems interacting with nuclear spins. We believe that these systems would be promising for microelectronic semiconductor applications that require integrated inductors.

## Acknowledgments

We acknowledge useful discussions with B. Pekerten, D. M. Couger, R. Salas. İ. A. is a member of the Science Academy- Bilim Akademisi–Turkey; A. M. B. thanks the Science Academy-Bilim Akademisi–Turkey for the use of their facilities.

**Author contributions**    A.M.B performed the Keldysh Green's function calculations with input from I.A. and wrote the manuscript with input from all authors. S.K. performed calculations for the inductive response of topological insulators and its entropic nature with input from all authors. A.B. and I.A. supervised the project.

**Funding information**    This research was supported by a Lockheed Martin Corporation Research Grant.

## A    Charge-spin coupled dynamics of the 3D topological insulator surface states

In this appendix, we show the spin sector of the quantum kinetic equation for the surface states of a 3D topological insulator. We start by performing the spin traces $\boldsymbol{\sigma}$ of Eq. (11) and obtain:

$$
\partial_t g_x + v_F \nabla_y g_0 + 2 v_F k_F \hat{k}_x g_z = -\frac{1}{\tau_0}\left[ g_x - \langle g_x \rangle - \hat{k}_y \langle g_0 \rangle \right]
$$
$$
-\frac{3}{\tau_{sf}}\left[ g_x + \frac{1}{3}\langle g_x \rangle + \frac{2}{3}\langle \boldsymbol{g} \rangle \cdot \boldsymbol{m} g_x - \frac{2}{3}\langle g_0 \rangle g_0 m_x + \frac{2}{3} m_x - \hat{k}_y \langle g_0 \rangle \right],
$$
(A.1)

for the $x-$ component, whereas the $\sigma_y$ trace yields:

$$
\partial_t g_y - v_F \nabla_x g_0 + 2 v_F k_F \hat{k}_y g_z = -\frac{1}{\tau_0}\left[ g_y - \langle g_y \rangle + \hat{k}_x \langle g_0 \rangle \right]
$$
$$
-\frac{3}{\tau_{sf}}\left[ g_y + \frac{1}{3}\langle g_y \rangle + \frac{2}{3}\langle \boldsymbol{g} \rangle \cdot \boldsymbol{m} g_y - \frac{2}{3}\langle g_0 \rangle g_0 m_y + \frac{2}{3} m_y + \hat{k}_x \langle g_0 \rangle \right],
$$
(A.2)

and finally $\sigma_z$ trace:

$$
\partial_t g_z + 2 v_F k_F \left( \hat{k}_x g_x + \hat{k}_y g_y \right) = -\frac{1}{\tau_0}\left[ g_z - \langle g_z \rangle \right]
$$
$$
-\frac{3}{\tau_{sf}}\left[ g_z + \frac{1}{3}\langle g_z \rangle + \frac{2}{3}\langle \boldsymbol{g} \rangle \cdot \boldsymbol{m} g_z - \frac{2}{3}\langle g_0 \rangle g_0 m_z + \frac{2}{3} m_z \right].
$$
(A.3)

Next, we assume that the quasiclassical Green's function is diagonal in the eigenstates of the 3D topological insulator Hamiltonian. We then insert the ansatz given in Eq. (23) and Eq. (24) and obtain the spin sector for the quantum kinetic equation:

$$\partial_t \langle \hat{k}_y g_0 \rangle + \frac{v_F}{2} \nabla_y \langle g_0 \rangle + \frac{\langle \hat{k}_y g_0 \rangle}{2\tau_0} = -\frac{3}{\tau_{sf}} \left( \frac{4}{3} \langle \hat{k}_y g_0 \rangle + \frac{2}{3} \left( \langle \hat{k}_y g_0 \rangle \boldsymbol{m} \cdot \langle (\hat{k} \times \hat{z}) g_0 \rangle \right) \right) - \frac{2}{3} \langle g_0 \rangle^2 m_x + \frac{2}{3} m_x \right),$$
(A.4)

and

$$\partial_t \langle \hat{k}_x g_0 \rangle + \frac{v_F}{2} \nabla_x \langle g_0 \rangle + \frac{\langle \hat{k}_x g_0 \rangle}{2\tau_0} = -\frac{3}{\tau_{sf}} \left( \frac{4}{3} \langle \hat{k}_x g_0 \rangle - \frac{2}{3} \left( \langle \hat{k}_x g_0 \rangle \boldsymbol{m} \cdot \langle (\hat{k} \times \hat{z}) g_0 \rangle \right) \right) + \frac{2}{3} \langle g_0 \rangle^2 m_y - \frac{2}{3} m_y \right).$$
(A.5)

Here, we keep the discussion to the kinetic equation for the in-plane electron spin polarization only. By using the generalized density matrix $F(\epsilon, \boldsymbol{R}) = n(\epsilon, \boldsymbol{R})/2\sigma_0 + \boldsymbol{s}(\epsilon, \boldsymbol{R}) \cdot \boldsymbol{\sigma}$, we arrive at the quantum kinetic equations for the spin sector given in Eq. (27) in the main text.

# B  Nuclear spin correlators

Our starting point is the two-point correlators for the nuclear spins, given in Eq. (15). We map the contour ordered nuclear spin correlations on the Keldysh space and get individual elements of the nuclear spin correlators:

$$D_{\alpha\beta}^{--}(1,2) = -i \left\langle \mathcal{T} \left( I_\alpha^{n_1}(t_1) I_\beta^{n_2}(t_2) \right) \right\rangle,$$
(B.1)

$$D_{\alpha\beta}^{-+}(1,2) = -i \langle I_\beta^{n_2}(t_2) I_\alpha^{n_1}(t_1) \rangle,$$
(B.2)

$$D_{\alpha\beta}^{+-}(1,2) = -i \langle I_\alpha^{n_1}(t_1) I_\beta^{n_2}(t_2) \rangle,$$
(B.3)

$$D_{\alpha\beta}^{++}(1,2) = -i \left\langle \tilde{\mathcal{T}} \left( I_\alpha^{n_1}(t_1) I_\beta^{n_2}(t_2) \right) \right\rangle,$$
(B.4)

where $\mathcal{T}(\tilde{\mathcal{T}})$ denotes the time ordering (anti-ordering) operator and we denote the position of the $i^{\text{th}}$ nuclear spin as $n_i$. Here, $D_{\mp(\pm)}$ are the lesser (greater) Green's function. Then, we calculate the expectation value of the nuclear spin operators, i.e. $\langle I_\alpha^{n_i} I_\beta^{n_j} \rangle = \delta_{n_1, n_2} \left( \delta_{\alpha\beta} + i\epsilon_{\alpha\beta\gamma} \langle I_\gamma^{n_1} \rangle - \langle I_\alpha^{n_1} \rangle \langle I_\beta^{n_1} \rangle \right)$. Taking the time ordering and anti-ordering into account, we have:

$$iD_{\alpha\beta}^{\pm\mp}(1,2) = \delta_{n_1, n_2} \left( \delta_{\alpha\beta} \pm i\epsilon_{\alpha\beta\gamma} m_\gamma^{n_1} - m_\alpha^{n_1} m_\beta^{n_2} \right),$$
(B.5)

$$iD_{\alpha\beta}^{\mp\mp}(1,2) = \delta_{n_1, n_2} \left( \delta_{\alpha\beta} \pm \text{sgn}(t_1 - t_2) i\epsilon_{\alpha\beta\gamma} m_\gamma^{n_1} - m_\alpha^{n_1} m_\beta^{n_2} \right),$$
(B.6)

where we define $m_\alpha^n \equiv \langle I_\alpha^n \rangle$. In the rest of the calculations, we consider only on-site correlations and assume low nuclear spin density. Performing a rotation in the Keldysh space as described in Ref. [22], we can then arrive at the retarded, advanced and Keldysh Green's function for the nuclear spins, as given in Eq. (15) in the main text.

Next, we calculate the self energy for electrons due to their interaction with the nuclear spins. We show the diagrammatic representation of the self energy due to nuclear spin scattering in Fig. 2(b). We consider two-point correlators for the nuclear spins only and obtain the

components of the self energy matrix $\underline{\Sigma}_m$ as:

$$\Sigma_m^{--}(\boldsymbol{R},\boldsymbol{k},t_1,t_2) = +i\frac{\lambda^2}{4}\int\frac{d^2\boldsymbol{k}'}{(2\pi)^2}\,\sigma_\alpha G^{--}(\boldsymbol{R},\boldsymbol{k}',t_1,t_2)\sigma_\beta\,D_{\alpha\beta}^{--}(\boldsymbol{R},\boldsymbol{k}-\boldsymbol{k}',t_1,t_2),$$

$$\Sigma_m^{++}(\boldsymbol{R},\boldsymbol{k},t_1,t_2) = +i\frac{\lambda^2}{4}\int\frac{d^2\boldsymbol{k}'}{(2\pi)^2}\,\sigma_\alpha G^{++}(\boldsymbol{R},\boldsymbol{k}',t_1,t_2)\sigma_\beta\,D_{\alpha\beta}^{++}(\boldsymbol{R},\boldsymbol{k}-\boldsymbol{k}',t_1,t_2),$$

$$\Sigma_m^{-+}(\boldsymbol{R},\boldsymbol{k},t_1,t_2) = -i\frac{\lambda^2}{4}\int\frac{d^2\boldsymbol{k}'}{(2\pi)^2}\,\sigma_\alpha G^{-+}(\boldsymbol{R},\boldsymbol{k}',t_1,t_2)\sigma_\beta\,D_{\alpha\beta}^{+-}(\boldsymbol{R},\boldsymbol{k}-\boldsymbol{k}',t_1,t_2),$$

$$\Sigma_m^{+-}(\boldsymbol{R},\boldsymbol{k},t_1,t_2) = -i\frac{\lambda^2}{4}\int\frac{d^2\boldsymbol{k}'}{(2\pi)^2}\,\sigma_\alpha G^{+-}(\boldsymbol{R},\boldsymbol{k}',t_1,t_2)\sigma_\beta\,D_{\alpha\beta}^{-+}(\boldsymbol{R},\boldsymbol{k}-\boldsymbol{k}',t_1,t_2). \tag{B.7}$$

We emphasize that the elements of the self energy matrix given in Eq. (B.7) is written in a different basis than the one we use in description of the electronic Green's function. To that end, we use the retarded, advanced and Keldysh representation of the self energy given in Eq. (B.7) in the main text.

## C  Nuclear spin polarization dynamics

In this appendix, we obtain the nuclear spin dynamics using the nonequilibrium Green's function method. We first obtain the self energy for the nuclear spin correlators as follows:

$$\Pi_{\alpha\beta}^{-+}(\boldsymbol{q},\Omega) = -i\frac{\lambda^2}{4}\int\frac{d^2\boldsymbol{k}}{(2\pi)^2}\int\frac{d\omega}{2\pi}\,\mathrm{Tr}\left[\sigma_\alpha G^{-+}(\boldsymbol{k},\omega)\sigma_\beta G^{+-}(\boldsymbol{k}-\boldsymbol{q},\omega-\Omega)\right]. \tag{C.1}$$

We then parameterize the electronic Green's function, namely $G \equiv G_\mu\sigma_\mu$ with $\mu = \{0,x,y,z\}$ and obtain:

$$\Pi_{\alpha\beta}^{-+} = -i\frac{\lambda^2}{4}\,\mathrm{Tr}\left[\sigma_\alpha\sigma_\gamma\sigma_\beta\sigma_\mu\right]\int\frac{d^2\boldsymbol{k}}{(2\pi)^2}\int\frac{d\omega}{2\pi}G_\mu^{-+}G_\gamma^{+-},$$

$$= -i\frac{\lambda^2}{2}\int\frac{d^2\boldsymbol{k}}{(2\pi)^2}\int\frac{d\omega}{2\pi}\left(\delta_{\alpha\beta}G_0^{-+}G_0^{+-} + i\epsilon_{\alpha\beta\mu}\left(G_\mu^{-+}G_0^{+-} - G_0^{-+}G_\mu^{+-}\right)\right.$$

$$\left. + G_\beta^{-+}G_\alpha^{+-} - \delta_{\alpha\beta}G_\gamma^{-+}G_\gamma^{+-} + G_\alpha^{-+}G_\beta^{+-}\right), \tag{C.2}$$

where we discard the momentum-energy variables $(\boldsymbol{q},\Omega)$ for clarity. Similarly, the greater component of the self energy is found to be:

$$\Pi_{\alpha\beta}^{+-} = -i\frac{\lambda^2}{4}\,\mathrm{Tr}\left[\sigma_\alpha\sigma_\gamma\sigma_\beta\sigma_\mu\right]\int\frac{d^2\boldsymbol{k}}{(2\pi)^2}\int\frac{d\omega}{2\pi}G_\mu^{+-}G_\gamma^{-+},$$

$$= -i\frac{\lambda^2}{2}\int\frac{d^2\boldsymbol{k}}{(2\pi)^2}\int\frac{d\omega}{2\pi}\left(\delta_{\alpha\beta}G_0^{+-}G_0^{-+} + i\epsilon_{\alpha\beta\mu}\left(G_\mu^{+-}G_0^{-+} - G_0^{+-}G_\mu^{-+}\right)\right.$$

$$\left. + G_\beta^{+-}G_\alpha^{-+} - \delta_{\alpha\beta}G_\gamma^{+-}G_\gamma^{-+} + G_\alpha^{+-}G_\beta^{-+}\right). \tag{C.3}$$

We evaluate the nuclear spin self energy by associating the lesser and greater Green's functions in the nuclear spin self energy with the distribution functions of the electrons. Similar to the electron spin dynamics, we make use of the generalized distribution matrix $F = \frac{n}{2}\sigma_0 + \boldsymbol{s}\cdot\boldsymbol{\sigma}$.

As an example, we focus on the first term in the integrand of Eq. (C.2), where we have:

$$\int \frac{d^2\boldsymbol{k}}{(2\pi)^2} \int \frac{d\omega}{2\pi} n(\boldsymbol{k})(1-n(\boldsymbol{k}-\boldsymbol{q}))\delta(\omega - \hbar v_F k + \mu)\delta(\omega - \Omega - \hbar v_F|\boldsymbol{k}-\boldsymbol{q}| + E_F)$$

$$= \int \frac{d^2\boldsymbol{k}}{(2\pi)^2} n(\boldsymbol{k})(1-n(\boldsymbol{k}-\boldsymbol{q}))\delta(\hbar v_F k - \Omega - \hbar v_F|\boldsymbol{k}-\boldsymbol{q}|), \tag{C.4}$$

where the $\delta$ function ensures the energy conservation due to scattering. We note that we obtain the other terms in the integrand of Eq. (C.2) in a similar fashion.

After obtaining the nuclear spin self energy, we now focus on the nuclear spin dynamics. In order to describe the dynamics of the nuclear spins, we first write down the kinetic equation for the lesser component of the nuclear spin correlators in the Wigner representation:

$$\dot{D}_{\alpha\beta}^{-+}(\boldsymbol{q},\Omega) = -\frac{i}{\hbar}\Big[\Pi_{\alpha\delta}^{-+}(\boldsymbol{q},\Omega)D_{\delta\beta}^{+-}(\boldsymbol{q},\Omega) - D_{\alpha\delta}^{-+}(\boldsymbol{q},\Omega)\Pi_{\delta\beta}^{+-}(\boldsymbol{q},\Omega)\Big], \tag{C.5}$$

where we omit the position and time variables for clarity.

As Eq. (C.5) is given in the Wigner representation, we need to integrate it over momentum $\boldsymbol{q}$ and energy $\Omega$ in order to obtain the dynamics of the nuclear spins. We achieve this by considering the equal time density matrix, which corresponds to integrating the overall quantum kinetic equation for the nuclear spins over $\Omega$. This allows us to set $\Omega = 0$ for Eq. (C.5). In that case, we have:

$$\int \frac{d^2\boldsymbol{q}}{(2\pi)^2} \dot{D}_{\alpha\beta}^{-+}(\boldsymbol{q},0) = -\frac{i}{\hbar} \int \frac{d^2\boldsymbol{q}}{(2\pi)^2}\Big[\Pi_{\alpha\delta}^{-+}(\boldsymbol{q},0)D_{\delta\beta}^{+-}(\boldsymbol{q},0) - D_{\alpha\delta}^{-+}(\boldsymbol{q},0)\Pi_{\delta\beta}^{+-}(\boldsymbol{q},0)\Big]. \tag{C.6}$$

We then assume that there exists only on-site correlations between the nuclear spin. Therefore, we ignore the momentum dependence of the nuclear spin correlators. We define $d_{\alpha\beta}^{-+} \equiv \int \frac{d^2\boldsymbol{q}}{(2\pi)^2} D_{\alpha\beta}^{-+}(\boldsymbol{q},0)$ and approximate the right hand side of Eq. (C.6) as:

$$\int \frac{d^2\boldsymbol{q}}{(2\pi)^2}\Big[\Pi_{\alpha\delta}^{-+}(\boldsymbol{q})D_{\delta\beta}^{+-}(\boldsymbol{q}) - D_{\alpha\delta}^{-+}(\boldsymbol{q})\Pi_{\delta\beta}^{+-}(\boldsymbol{q})\Big] \approx \Big(\pi_{\alpha\delta}^{-+}d_{\delta\beta}^{+-} - d_{\alpha\delta}^{-+}\pi_{\delta\beta}^{+-}\Big), \tag{C.7}$$

where we define $\int \frac{d^2\boldsymbol{q}}{2(\pi)^2} \Pi_{\alpha\delta}^{-+}(\boldsymbol{q},0) \equiv \pi_{\alpha\delta}^{-+}$. In this case, we obtain:

$$\dot{d}_{\alpha\beta}^{-+}(\boldsymbol{r},t) = -\frac{i}{\hbar}\Big(\pi_{\alpha\delta}^{-+}d_{\delta\beta}^{+-}(\boldsymbol{r},t) - d_{\alpha\delta}^{-+}(\boldsymbol{r},t)\pi_{\delta\beta}^{+-}\Big), \tag{C.8}$$

as also given in Eq. (35). We note the self energy obeys the following relation, $\pi_{\alpha\beta}^{-+} = \pi_{\beta\alpha}^{+-}$. This proves that the kinetic equation (Eq. (35)) for all the diagonal components of the nuclear spins correlators is trivial, $\dot{d}_{\alpha\alpha} = 0$, as expected.

Owing to the assumption we use, we only need to integrate the nuclear spin self energy over the momentum. We again exemplify this for the first term in the self energy and integrate Eq. (C.4) over momentum $\boldsymbol{q}$ and we have:

$$= \int \frac{dk}{2\pi} 2k^2 \int_0^{2\pi} \frac{d\theta_k}{2\pi} \int_{-\pi/2+\theta_k}^{\pi/2+\theta_k} \frac{d\theta_q}{2\pi} \cos(\theta_k - \theta_q)n(k,\theta_k)(1-n(k,\theta_k-\theta_q))$$

$$= \int \frac{dk}{2\pi} 2k^2 \int_0^{2\pi} \frac{d\theta_k}{2\pi} n(k,\theta_k)\Big[\frac{1}{\pi} - \int_{-\pi/2+\theta_k}^{\pi/2+\theta_k} \frac{d\theta_q}{2\pi} \cos(\theta_k - \theta_q)n(k,\theta_k-\theta_q))\Big]$$

$$\approx \frac{2}{\pi} \int \frac{dk}{2\pi} k^2 \int_0^{2\pi} \frac{d\theta_k}{2\pi} n(k,\theta_k)(1-\langle n(k)\rangle)$$

$$\approx \frac{2}{\pi} \int \frac{dk}{2\pi} k^2 \langle n(k)\rangle (1-\langle n(k)\rangle), \tag{C.9}$$

where $\langle n(k) \rangle$ denotes the angular average of the occupation factor, $q = |\boldsymbol{q}|$ and $k = |\boldsymbol{k}|$. We note that we use the condition $q = 2k \cos(\theta_k - \theta_q)$ due to the energy conservation. The remaining terms in the nuclear spin self energy can be evaluated similarly.

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
