# Peer review of "Entropy Driven Inductive Response of Topological Insulators"

_SciPost Physics Core, doi:SciPost Phys. Core 8, 023 (2025)_

## Round 1 · Referee Report · Anonymous (Referee 1) · 2024-7-17

Report
In this manuscript, the authors investigate theoretically the charge and spin dynamics of the surface states of a disordered three-dimensional topological insulator, including the Fermi contact hyperfine interaction with the nuclear spins of the host material. They employ the Keldysh formalism within a quasiclassical approach, introducing the coupling of the electrons to (nonmagnetic) impurities and the nuclear spins to lowest order. This allows to derive the quantum kinetic equations describing the coupled dynamics of the electronic spin and charge and the nuclear spins. These equations can then be reduced to Boltzmann-like equations for the charge and spin dynamics, from which simple transport equations follow.
The authors find that the spin-momentum locking of the electrons on the surface yields a direct coupling between the nuclear spin dynamics and the surface charge currents. They show that in a transport setup this results in (i) a small correction to the conductance of the surface states, due to scattering off nuclear spins and (ii) a small contribution to the current driven by the entropy of the nuclear spin system. The latter contribution can be seen as a small inductive contribution to the impedance of the surface, the magnitude of which depends directly on the geometry of the device. This could potentially lead to applications in the form of small and controllable inductive elements. Finally, the authors estimate the magnitude of the effect in a few candidate materials, confirming that it will indeed probably be on the small side.
The manuscript is very well written and easy to follow: Sections 2.1-2.5 give a very readable and pedagogic overview of the calculation, making it possible to follow all steps in detail. Although there is a large overlap of ideas between this work and Ref. [1] and the estimated magnitude of the resulting inductive effects is somewhat disappointing, I think that the manuscript is timely, interesting and of value for the community. Especially the very clearly worked out interpretation of the effect of the nuclear spin dynamics in terms of an inductive contribution is interesting and does provide a novel and synergetic link between different concepts, to my taste. Therefore, altogether I recommend publication of this work in SciPost. I do have a few comments which I think the authors could address first:
- The Fermi contact hyperfine interaction as written in Eq. (2) is directly proportional to the weight of the electronic wave functions at the position of the nuclei. For the usual s-type states in the conduction band of most semiconductors this will indeed be by far the dominating contribution to the interaction. For p-type states, this contact interaction will in principle be absent and the interaction will be much smaller and can be qualitatively different. I am not an expert on the band structure of real-life topological insulators and the resulting orbital structure of the surface states, but I imagine that they are not of (pure) s-type. This issue is addressed in Sec. 3.2 by a statement that the interaction will be reduced because of this, with a reference to a PhD thesis I cannot access. It would be good if the authors could address this slightly more thoroughly: Why is the resulting interaction expected to be of the Heisenberg type, and not, e.g., dominated by Ising terms, depending on the detailed orbital angular momentum of the states? Is the behavior of all materials investigated in Sec. 3.2 expected to be the same in this respect? Etc.
- The authors write below Eq. (44) that the entropy-induced current results from the fact that a polarized nuclear spin system prefers to raise its entropy, which "can only be achieved by transferring their spin angular momentum to electron spins via hyperfine interaction." In reality, such entropy gain will also be achieved via nuclear spin diffusion into the bulk (through nuclear spin-spin interactions). Typically, this is a relatively slow process, but since the effect on the charge current found here is also very small it would be good to compare the relevant time scales more quantitatively.
- Since I consider the pedagogic style of the manuscript one of its key values, I would recommend adding one initial step at the beginning of App. B, defining the greater and lesser Green functions and showing how (67,68) follow. This would balance that part with the very detailed introduction provided around Eqs. (3-7).
Recommendation
Publish (meets expectations and criteria for this Journal)

---

## Round 2 · Referee Report · Anonymous (Referee 1) · 2024-11-22

Report

I read the new version of the manuscript as well as the authors' reply to my previous comments, and I think that the manuscript is ready for publication now.

Concerning point (2), the reply is not fully complete (in my view): Comparing the efficiency of two processes by comparing two quantities that have different dimensions is of course questionable; the in-plane wave functions and densities should also be included to make a fair comparison. I, however, do (and did) agree that the role of nuclear spin diffusion will be small and I am fine with disregarding it in this context, I was just interested in a quantitative comparison.

Recommendation

Publish (meets expectations and criteria for this Journal)

---

## Round 2 · Author Response

We thank the referee for their comments and overall positive evaluation. Here, we give our point by point response to address referee's comments and questions:

1) "Why is the resulting interaction expected to be of the Heisenberg type, and not, e.g., dominated by Ising terms, depending on the detailed orbital angular momentum of the states? Is the behavior of all materials investigated in Sec. 3.2 expected to be the same in this respect? Etc."

  • We thank the referee for their comment. In this manuscript, we focus on the Fermi contact interaction, which is the dominant interaction for electrons in s-bands. The orbital structure of topological surface states typically consists of a mixture of s- and p-type orbitals, and their exact relative weight varies across different materials. For electrons in p-bands, as the referee points out, the hyperfine interaction can indeed be more complicated. However, the energy scale of the hyperfine interaction for electrons in p-type orbitals is at least an order of magnitude smaller than that for electrons in s-type orbitals. Hence even if the bands mix, as long as the s-orbital component is significant, the dominant interaction will be the Fermi contact interaction. This was noted in a previous study that focused on the effective hyperfine interaction for the helical edge states in HgTe, which are a mixture of p-type and s-type orbitals, see. In the present work, for the sake of simplicity, we ignore the subleading contribution and assume that the electron/nuclear spin dynamics is dominated by the the contribution of the s-type orbitals. The effective interaction strength is determined by the weight of the s-band in the surface state orbital structure, multiplied by their bare interaction strength. This approximation is valid for materials with significant s-type orbital character, even if it is smaller than the p-orbital weight. To make our point more accessible to the reader, we have added the reference we mentioned above and two more references.

2) "..In reality, such entropy gain will also be achieved via nuclear spin diffusion into the bulk (through nuclear spin-spin interactions). Typically, this is a relatively slow process, but since the effect on the charge current found here is also very small it would be good to compare the relevant time scales more quantitatively."

* The referee is right in pointing out that spin-flip interactions via hyperfine coupling is not the only mechanism causing a change in nuclear polarization: dipole-dipole interaction between nuclear spins leads to a diffusion of the nuclear spin (polarized by the topological surface state) into the bulk of the material. This mechanism changes the entropy of the nuclear spin subsystem as well. However, the energy scale of the dipole-dipole interaction is orders of magnitude lower than the Fermi contact interaction. As an example, we consider two Bismuth nuclear spins ($\gamma_{Bi}$: 6.9 MHz/T, [see Ref.](https://doi.org/10.61092/iaea.yjpc-cns6), spaced apart by the lattice constant of BST $a_0 =$ 0.439 nm. The resulting dipole-dipole interaction [given in Ref.](https://doi.org/10.1016/S0167-6881(98)80007-4) is proportional to $\mu_0 \gamma_1 \gamma_2 \hbar^2/(4\pi r^3) =$ 8.310$^{-14}$ eV. In comparison, the Fermi contact interaction is proportional to $\lambda = A_0 v_0/\xi =$ 2.5 $\mu$eV nm$^2$, which is multiple orders of magnitude larger than the dipole-dipole interaction. Furthermore, we point out that the orders of magnitude difference between Fermi contact interaction and dipole-dipole interaction is quite generic. Therefore we assume that entropy loss via diffusion through dipole-dipole interaction is negligible on the timescales of the inductive effect.

3) "Since I consider the pedagogic style of the manuscript one of its key values, I would recommend adding one initial step at the beginning of App. B, defining the greater and lesser Green functions and showing how (67,68) follow. This would balance that part with the very detailed introduction provided around Eqs. (3-7)."

  • We thank the referee for their comment. Following their suggestion, we have extended the discussion around Eq. 67 and 68, showing how to get the lesser and greater Green's functions and calculate them.

---

## Round 2 · List of Changes

* We added two more references about the effect of s- and p-orbital mixing on the hyperfine coupling in Section 3.2. We also made Ref. 26 accessible.
* Following referee's suggestion, we extended the discussion about the in Appendix B.
* We fixed a couple of typos.

---

## Editorial Decision

published